# Metabolic Profiling of Wheat Seedlings Under Oxygen Deficiency and Subsequent Reaeration Conditions

**DOI:** 10.3390/ijms262311610

**Published:** 2025-11-30

**Authors:** Vladislav V. Yemelyanov, Roman K. Puzanskiy, Ekaterina M. Bogdanova, Sergey A. Vanisov, Maksim D. Dubrovskiy, Victor V. Lastochkin, Anastasia A. Kirpichnikova, Alla N. Brykova, Alexey L. Shavarda, Maria F. Shishova

**Affiliations:** 1Department of Genetics and Biotechnology, Faculty of Biology, St. Petersburg State University, Universitetskaya em., 7/9, 199034 St. Petersburg, Russia; puzansky@yandex.ru (R.K.P.);; 2Laboratory of Analytical Phytochemistry, V.L. Komarov Botanical Institute of the Russian Academy of Sciences, St. Professora Popova, 2, 197376 St. Petersburg, Russia; 3Center for Molecular and Cell Technologies, Research Park, St. Petersburg State University, 199034 St. Petersburg, Russia; 4Department of Plant Physiology and Biochemistry, Faculty of Biology, St. Petersburg State University, Universitetskaya em., 7/9, 199034 St. Petersburg, Russiamshishova@mail.ru (M.F.S.); 5Department of Genetic Resources of Wheat, N.I. Vavilov All-Russian Institute of Plant Genetic Resources, Bolshaya Morskaya St., 42-44, 190000 St. Petersburg, Russia

**Keywords:** anoxia, reoxygenation, metabolomics, hydroxyl carboxylic acids, adaptation, tolerance, *Triticum aestivum*

## Abstract

The ability of plants to survive oxygen deficiency is associated with significant changes in metabolism. Metabolic profiling of wheat seedlings under anoxia and subsequent reoxygenation conditions was performed using GC-MS. A total of 374 and 298 compounds were detected in root and shoot metabolomes, respectively. All intermediates of central metabolism were identified. Early anoxic responses of root and shoot metabolomes showed similarity, leading to the accumulation of amino acids (Ala, GABA and Tyr), carboxylates (lactate and succinate), nucleotides and amines, together with a decrease in sugars. The metabolic response to long-term anoxia varied significantly in the roots and shoots of wheat seedlings and was related to the redistribution of carbon flux from glycolysis predominantly to lipids in the roots, while it was directed to carboxylates and GABA in the shoots. Imposition of 24 h of reaeration after short-term anoxia (6 h) switched the metabolome toward a normoxic profile, predominantly in roots. Anaerobically down-regulated metabolites were accumulated, while anaerobic intermediates were depleted post-anoxia. The effects of more prolonged anoxia on wheat seedling metabolomes were less reversible, particularly in shoots. Interestingly, several metabolites with not fully understood roles (e.g., hydroxyl carboxylates, α,ω-dicarboxylic acids, polyols) were detected under anoxic conditions in wheat seedlings, which could potentially serve as markers of plant sensitivity to oxygen deficiency.

## 1. Introduction

Considering the intensive climate change occurring over recent decades, the importance of environmental challenges such as oxygen deprivation has been revealed. Oxygen deprivation occurs at different intensities during partial or complete flooding due to local intensive rainfall, poor drainage, snow, ice covering, asphalt pavement and so on [1,2,3]. Plants can survive oxygen-deficient conditions via two adaptation strategies. The first is to escape the oxygen-deficient environment (low-oxygen escape syndrome, LOES). This strategy entails acceleration of shoot growth, stimulation of adventitious root formation, underwater photosynthesis and the development of a number of anatomic changes like aerenchyma or waxy cuticles [1,2,4,5,6,7]. This has been observed in most hydrophytes, including deepwater rice (a specific ecotype of *Oryza sativa*), *Ranunculus sceleratus*, *Rorippa amphibia* and *Rumex palustris* [8,9,10,11,12]. The alternative strategy is called the quiescence strategy (low-oxygen quiescence syndrome, LOQS), which is typically used by Indian *Sub1* rice varieties, *Arabidopsis thaliana*, *Oenathe aquatic*, *R. sylvestris* and so on [9,10]. LOQS involves a decrease in growth rate and is based on different metabolic adaptations aimed at preventing energy starvation, the formation of toxic fermentation products and post-anoxic oxidative damage [1,6,13]. The necessity to maintain metabolism under stressful conditions, such as those imposing a significant limitation in energy supply (respiration shortage and photosynthesis impairment), is a feature common to both strategies.

The first data on the balance of different groups of metabolites under hypoxic/anoxic conditions were obtained using conventional biochemistry methods and have been recently confirmed through metabolic profiling based on the application of several technological platforms—including gas- or high-performance liquid chromatography, capillary electrophoresis coupled with mass spectrometry and NMR detection—collectively referred to as “omics” methods [14,15,16,17,18,19,20]. These methods are advantageous in analyzing complex alterations in a wide range of low-molecular-weight compounds mostly involved in primary metabolism. The results of those investigations have elucidated the involvement of glycolysis, fermentation, the Krebs cycle and the metabolism of amino acids in the metabolic adjustment in response to oxygen deprivation [2,4,5,14,15,16,17,18,19,20,21,22,23].

Nevertheless, the number of metabolomics investigations focused on hypoxia/anoxia is limited compared to other environmental issues [24,25], and these processes have mostly been assessed in *A. thaliana*, as a well-characterized model plant [15,17]. Another set of model objects includes plants known for their tolerance to oxygen deprivation. Wetland species such as *Potamogeton anguillanus* [19], *Zostera marina* [26], *Epilobium hirsutum* and *E. palustre* [27], *Ranunculus* species [28], as well as crop plants like rice [16,18,20], have been studied.

In most cases, oxygen limitation triggered elevated levels of soluble sugars, pyruvate, succinate and lactate in plant tissues [25]. Amino acids related to glycolysis, including the phosphoglycerate family (Ser and Gly), the shikimate family (Phe, Tyr and Trp) and the pyruvate family (Ala, Leu and Val), were accumulated. Members of the Asp family (Asn, Lys, Met, Thr and Ile) and the Glu family (Glu, Pro, Arg and γ-aminobutyric acid [GABA]) were also accumulated and are known to link glycolysis with the altered Krebs cycle and to provide alternative pathways of NAD(P)H reoxidation, avoiding excessive accumulation of toxic fermentation products (lactate, acetaldehyde, ethanol) [2,5,20,21,23,25].

An intriguing question is whether it is possible to distinguish any differences in metabolic adaptation between flood-tolerant and flood-sensitive plants. Wheat (*Triticum aestivum*) belongs to the sensitive group and is commonly used as a counterpoint to the resistant rice plant [2,16,29]. The metabolic profiling studies carried out with wheat under hypoxic/anoxic conditions could be counted on the fingers of one hand.

In *T. aestivum* seedlings, 1 day of anoxic treatment triggered a decrease in soluble sugar and sugar phosphate levels, while no changes in 3-phosphoglycerate were detected in coleoptiles [16]. At the same time, the effects of anoxia on the levels of most of the Krebs cycle metabolites were rather negative, with the exception of succinate, which was found to be accumulated. Amino acid contents were altered differently: levels of Ala, Arg, Asn, Glu, Gln, Ile, Leu, Met, Thr, Trp, Tyr and Val were unchanged; Asp and Ser were decreased; and Gly, Lys, Phe, Pro and GABA were elevated.

A very interesting comparison was provided using two cultivars of *T. aestivum* that differed in tolerance to hypoxia (intolerant Frument and tolerant Jackson). It was shown that flooding (up to 16 days) led to a decrease in sucrose, fructose and glucose content in shoots in both genotypes [30]. Metabolome analysis revealed the difference between the cultivars on the 12th day of oxygen deprivation. Higher accumulation of amino acids was found in Frument than in Jackson. Prolongation of hypoxia led to a gradual decrease in the amino acid level in Frument and an increase in Jackson (Leu, Lys, Met, Phe, Pro, Thr, Trp, Tyr and Val). The depletion of the amino acid pool by the 16th day of stressor application was assumed to be common to the reaction of both cultivars due to the general damage to plants. GABA was accumulated only in less tolerant cultivars, and organic acids were not analyzed [30].

The list of examples would not be complete without extended investigation of the metabolomes of several wheat cultivars differing in their sensitivity to hypoxia. A previous study evaluated cultivars at different temperatures from 28 to 15 °C [31]. Sugar profiles of coleoptiles and roots of both the most tolerant Calingiri and least tolerant Ducula cultivars decreased at all tested temperatures during anoxia (1 day). The higher temperature had a severe effect. In contrast, the fructose level was elevated in Ducula coleoptiles at both 28 and 20 °C. Tested cultivars showed different dynamics of glucose-6-phosphate and fructose-6-phosphate, especially in roots, under this range of temperatures. In general, an increase in temperature led to a greater reduction in the levels of glucose-6-phosphate and fructose-6-phosphate, but there was no direct similarity with the alteration in the levels of soluble sugars. Likewise, the effect of anoxia on carboxylic acids was more pronounced at higher temperature in both the coleoptiles and roots of three cultivars, including Ducula. The effect was mainly negative. However, succinate content was elevated in roots of four of the five tested cultivars (except Ducula) at higher growth temperature. Colder conditions (15 °C) led to accumulation of this acid in two cultivars (Calingiri and Carnamah). Regarding amino acids, it was estimated that a large number of those metabolites showed an increase at 15 °C compared to 28 °C. The most significant changes were shown for Pro, Ala and GABA. A pronounced depletion was revealed for Asp, Glu, Gln and oxoproline. Similar trends were observed in the roots, where the accumulation of amino acids correlated with anoxia tolerance at low temperatures. Tissue- and temperature-dependent metabolic adaptations to anoxia were revealed [31]. Taken together, the available data demonstrate that wheat plants have some specificity in anoxic metabolic profiles, mostly concerning sugar and amino acid metabolism. However, further investigations are absolutely necessary.

Special interest is expected to be directed toward the metabolic changes occurring under post-anoxic reoxygenation. This phenomenon was found to be important in animals, including humans, because of its cardiological and neurological damage [32]. The major challenges in planting during this period concern post-anoxic oxidative damage, dehydration and vulnerability to phytopathogens and phytophages [6,33,34]. However, little is known about the metabolic profiles of plant tissues in this context. Some data revealed that even a short period of waterlogging (3 days) in 3-week-old wheat plants caused a severe effect on growth during further recovery periods. This shortage of growth correlated with an alteration in N and C concentrations [35]. Thus, this effect could alter metabolic adaptations.

The aim of our study was to evaluate the metabolic alterations induced by gradually developing anoxia in roots and shoots of wheat seedlings. Special focus was placed on the metabolic recovery following the post-anoxic period. The GC-MS approach was used for metabolic profiling.

## 2. Results

### 2.1. General Characteristics of Metabolic Profiles Under Anoxic Conditions

Metabolic profiling of wheat seedling roots revealed 374 compounds, 174 of which were annotated (118 as individual compounds and 56 as chemical classes, Appendix A). Among them were amino acids (29 in total, 20 proteinogenic), other nitrogenous compounds (14), carboxylic acids (15, including 5 from the Krebs cycle), fatty acids and derivatives (21), as well as 7 terpenoids and 5 phenolic compounds. In the obtained root metabolite profiles, sugars were the most abundant (69). Among them were 18 pentoses, 17 hexoses and 34 complex sugars, as well as polyols (10) and sugar oxidation acids (5, including ascorbate). The metabolome of wheat shoots was revealed to be less numerous and consisted of 298 metabolites, of which 154 were annotated (100 as individual compounds and 54 as chemical classes, Appendix A). These included 16 amino acids (10 proteinogenic); 17 carboxylic acids, including 5 Krebs cycle intermediates, 5 fatty acids and their derivatives, as well as 12 nitrogen-containing metabolites (nitrogenous bases, amines and di- and polyamines); sugar alcohols (9); acids of primary oxidation of sugars (6) and terpenoids (5). There were slightly more carbohydrates in the shoot metabolome (73) than in the roots, including pentoses (12), hexoses (25) and oligosaccharides (36). The contents of oligosaccharides accounted for 49% of the total pool of analyzed carbohydrates in both roots and shoots of wheat seedlings.

To compare metabolomes under different aeration regimes, we present them in a lower-dimensional space revealed through principal component analysis (PCA). Each of the metabolite profiles, numbering about 300–400 metabolites, is represented as a point (Figure 1). The proximity of points reflects the similarity of patterns of metabolite accumulation. Temporal alterations in roots metabolomes under anoxic and normoxic conditions were orthogonal and associated with PC1 (41.4%) and PC2 (12.7%), respectively (Figure 1a). In both normoxia and anoxia, gradual changes to metabolite profiles occurred in darkness, which accelerated after 24–72 h of exposure. In turn, PC3 (7.2%) reflected early metabolome alterations both under anoxic and normoxic conditions, probably induced by darkness (Figure 1b). In addition, PC3 reflected metabolic rearrangements during late anoxia; this shift was the opposite of the initial one. Increasing the anaerobic exposure time resulted in more pronounced segregation of the metabolomes. Only after 6–12 h of anoxia did the metabolomes of the experimental and control normoxic variants differ significantly. It can be assumed that in the initial stages of the experiment, in addition to anoxia, the lack of illumination also had a significant effect.

Similar changes occurred in wheat shoots. As with the roots, the metabolomic effects of prolonged anoxia were related to PC1 (29.8%) (Figure 1c). Under anoxic conditions, shoot metabolomes began to differ clearer after 12–24 h of exposure. In turn, the effects of early anoxia were associated with PC3 (10%) (Figure 1d). The effects of anoxia contained in PC3 were again reversed at later stages. Unlike the roots, the shoots did not demonstrate clear metabolomic differences during the first 24 h of normoxia in the dark. The absence of illumination in a normoxic environment led to significant metabolic differences by 72 h.

In summary, both roots and shoots showed multicomponent and nonlinear changes during anoxia. This is well illustrated by the graphs of PC1 and PC3 scores (Figure 1b,d). If the trend along PC1 was linear, then the dynamics along PC3 consisted of two trends. The first was before 12 h, and the second reverse trend started after 12 h of anoxic exposure.

### 2.2. Metabolic Profiles Under Short-Term Anoxic Conditions (1–3 h)

Changes in metabolic profiles of wheat roots were relatively weak in the first three hours of anoxia. However, the profiles of anoxic plants were clearly separated from the profiles of pre-anoxic plants along PC1 (Figure 2a). To identify metabolites with dynamics associated with anoxia in the first 3 h, we built an Orthogonal Projections to Latent Structures (OPLS) model. The model included one orthogonal component (Q^2^ = 0.7, *p* ≤ 0.001). The predictive component accounted for 19% of the variance. Figure 2c shows a bar plot of OPLS model loadings with VIP > 1, which is combined with a heatmap of average normalized content values in wheat roots. As can be seen, anoxia in the initial hours led to a drop in the levels of sugars, including oligosaccharides, sucrose, glucose, fructose and hexose phosphates along with the Krebs cycle carboxylates (aconitate and malate), amino acids (Asp, Glu and oxoproline), saturated fatty acids and their derivatives, as well as phenolics (arbutin). On the other hand, there was an accumulation of glycerol and its phosphate; a limited pool of pentoses; and a number of carboxylates, including pyruvate, lactate, succinate, 2-hydroxyglutarate and 3- and 4-hydroxybutyrate (hydroxybutanoate). Levels of amino acids, including α- and β-Ala, GABA, Gly, Pro, Phe, Tyr and other nitrogenous compounds (amines [ethanolamine and 1,3-diaminopropane] and nitrogenous bases [uracil, urate, xanthine]), as well as fatty acids and their derivatives, were also elevated (Figure 2c, Appendix A).

As in the case of roots, changes in metabolic profiles of wheat shoots were weak in the first three hours of anoxic exposure. As shown in Figure 2b, the profiles of anoxic plants differed from the pre-anoxic ones along PC1. The OPLS model for the first three hours included one orthogonal component (Q^2^ = 0.85, *p* ≤ 0.001). The predictive component accounted for 19% of the variance, as in the case of roots. The alteration in the shoot metabolic profile was very similar to that in roots of wheat seedlings (Figure 2d, Appendix A). The levels of sugars, sugar phosphates and amino acids (Asp, Glu and oxoproline) were down-regulated, while carboxylates (glycerate, lactate, succinate and 4-hydroxybutirate), amino acids (Ala, Cys, GABA and Tyr) and nitrogenous bases (adenine, allantoin, guanine and uracil/uridine) were accumulated.

We constructed an SUS plot to compare the effects of short-term anoxia on wheat root and shoot metabolomes, scattering the identified metabolites in the space of the loadings from corresponding to the OPLS models (Figure 3a). Some similarity (rho = 0.43, *p* = 0.0003, rho—Spearman’s correlation) was observed. A particularly striking common aspect was the increase in the availability of Ala, GABA, succinate, lactate, Tyr, uracil and amines together with a decrease in Asp, Glu and sugars during the initial hours of anoxia. Nevertheless, there were several differences, mainly concerning a higher intensity of changes and a greater number of compounds (amino, fatty and hydroxyl acids) accumulated in the roots.

Figure 3b presents the results of metabolite set enrichment analysis (MSEA) showing the extent to which biochemical pathways are affected by the factors. Short-term anoxia stimulated the metabolism of pyruvate, α- and β-Ala and some other amino acids (derived from pyruvate, shikimate, Asp and Glu) and carboxylates (propionate and butyrate). The accumulation of metabolites involved in the metabolism of sucrose and starch as well as Arg and His was repressed. An increase in the level of sterol metabolism was also observed in roots. In many cases, more pronounced effects in roots are observed.

### 2.3. Metabolic Profiles Under 6 h and Long-Term Anoxic Conditions (12–72 h)

#### 2.3.1. Metabolic Profiles of Wheat Roots Under Long-Term Anoxic Conditions

Starting from 6 h of treatment, the differences between control and anoxic plant metabolomes became more distinct (Figure 1). We conducted pairwise comparisons of metabolic profiles after anoxia with similar normoxic ones. Four OPLS-DA models were built. The predictive components were associated with 30, 48, 58 and 64% of the variance for 6, 12, 24 and 72 h of anoxia, respectively. Figure 4 is a heatmap showing the Fold Change (FC) differences between root metabolomes of control (normoxic) and anoxic plants. Asterisks indicate cases with VIP > 1. The levels of some pentoses, sucrose and a limited number of complex sugars, hexose phosphates (including glucose-6-phosphate and fructose-6-phosphate), *myo*-inositol and ascorbate at 6 h of anoxia continued to decrease, as was the case during the imposition of anaerobic conditions. In contrast, the majority of the sugar pool returned to the control level, including glucose and fructose. Moreover, some pentoses and oligosaccharides were accumulated (Figure 4, Appendix A). Carboxylic acids showed different trends. Pyruvate, lactate, succinate, 2-hydroxyglutarate and hydroxybutyrates, which were elevated at the first hour of anoxia, were depleted to control or below control levels up to 6 h, while citrate and malate were accumulated. The pool of amino acids (α- and β-Ala, Gly, Glu, Ile, Leu, oxoproline, Pro, Phe, Tyr, Trp, Ser and Val) was down-regulated to normoxic or even lower levels; in this case, only Asp was accumulated. The abundance of other nitrogenous compounds (amines and nitrogenous bases/nucleosides), as well as of fatty acids and their derivatives, continued to be elevated.

The transition from 6 to 12 h of anoxia and then to 24 and 72 h aggravated the changes in metabolic profiles that had begun after 6 h. Long-term anoxia depleted levels of the majority of pentoses, hexoses and hexose phosphates; oligosaccharides (including sucrose); as well as sugar alcohols and sugar acids (Figure 4). Nonetheless, some pentoses and complex sugars were accumulated up to 24 h of anoxia. Most carboxylates were down-regulated by long-term anaerobic treatment in wheat roots, including pyruvate, succinate, fumarate, malate, shikimate, quinate, 3-hydroxybutyrate and 2-hydroxyglutarate. The abundance of 3-hydroxypropionate and 4-hydroxybutyrate was decreased at 6–12 h of anoxia and then elevated by 24–72 h of treatment. On the contrary, the citrate level was increased at 6–12 h of anoxia and shortened by 24–72 h. Aconitate was up-regulated throughout long-term anoxia (12–72 h). The pool of amino acids started to be depleted after short-term anoxia (3–6 h) and was exhausted during long-term anaerobic treatment (up to 72 h). An interesting exception was Asp, Cys and ornithine, the levels of which were higher at 12–24 h of oxygen deprivation. Nitrogenous bases and nucleosides showed an increase throughout long-term treatment, and amines were depleted after 6–12 h of oxygen deprivation. Glycerol (but not its phosphates) and 2,3-butanediol were accumulated under such conditions. The characteristic changes in the prolonged anoxic metabolome in wheat roots also concerned lipid metabolism. The increase in the levels of a large number of fatty acids (including polyunsaturated ones), their derivatives and sterols were noticeable, mainly starting from 12 h of anoxia (Figure 4).

The enrichment analysis confirmed that the main response of roots to long-term anoxia was the increase in the levels of fatty acid and steroid metabolism (Figure 5). Moreover, this effect increased over time. Nucleotide metabolism was also activated. At the same time, metabolism of starch, sucrose and hexoses, as well as different amino acids, was inhibited. Pathways of propionate and butyrate metabolism initially activated in the initial hours of anoxic treatment were down-regulated by prolonged oxygen deprivation.

#### 2.3.2. Wheat Shoot Metabolic Profiles Under Long-Term Anoxic Conditions

Significant changes also occurred in wheat shoot metabolomes after 6 h of anoxia, but the trends that emerged after 3 h of exposure were generally maintained. The same tendencies of alterations continued during prolonged anoxia (12–72 h) (Figure 1c,d). The predictive components of the four OPLS-DA models characterizing the metabolic effects of anoxia accounted for 32, 48, 34 and 45% of the variance for 6, 12, 24 and 72 h, respectively. Q^2^ was between 0.87 and 0.99. The alteration in the shoot metabolic profile was slightly different to that obtained in roots of wheat seedlings (Figure 6, Appendix A). The levels of sucrose, some other complex sugars, hexoses and hexose phosphates continued to decrease at 6 h and were down-regulated throughout anoxic treatment (up to 72 h). Contrarily, the abundance of fructose and galactose showed an increase along with pentoses and a number of oligosaccharides, which started to be accumulated upon short-term oxygen limitation. The level of ascorbate was decreased, while other sugar acids, including gluconate, were up-regulated. Polyols, including *myo*-inositol, glycerol and 2,3-butanediol, were accumulated, particularly after 12–24 h of anoxia. A number of carboxylic acids showed greater accumulation under long-term anoxic conditions. Among them were glycerate, lactate, succinate, fumarate, glycolate, shikimate and 4-hydroxybutyrate. The 3-hydroxypropionate level was elevated up to 24 h of anoxia, while citrate and aconitate were transiently accumulated at 12 h of treatment. On the contrary, pyruvate, malate, malonate and oxalate showed a decrease. Amino acids demonstrated multidirectional, often non-monotonic trends. One can highlight an increase in GABA in the first 24 h and the transient elevation in Gly, Lys and Tyr, while the vast majority of amino acids (Asp, Asn, Glu, His, Met, oxoproline and Ser) were down-regulated throughout the entire period of long-term anoxia. The level of Gln was high only at 72 h of anoxia. α-Ala was detected in the metabolic profile with a decreasing trend, but differences were insignificant (Appendix A). Nitrogenous bases and nucleosides showed a dramatic increase throughout long-term treatment in shoots together with a small number of fatty acids, while a decrease was more typical for acylglycerols and steroids.

MSEA revealed that, despite the decrease in the abundance of pyruvate, the levels of other intermediates of its metabolism were up-regulated (Figure 7). Plants after 6 h of anoxia were characterized by a slight increase in glycolysis and the oxidative pentose phosphate pathway and a decrease in amino acid metabolism, and 12 h of oxygen deprivation led to stimulation of the components of the Krebs cycle. Prolonged anoxia (12–72 h) contributed to the accumulation of intermediates of metabolism of nitrogenous bases and nucleotides, propionate and butyrate metabolism, and repression of steroid metabolism. Lipid metabolism was stimulated to a lesser extent than in the roots. The staged changes were less pronounced, and the shoot metabolic profile gradually transformed from 3 to 72 h of anaerobic exposure.

We plotted the SUS plot of the loadings of identified metabolites corresponding to the OPLS-DA models (Appendix A) to compare the effects of long-term anoxia on wheat root and shoot metabolomes. There was practically no relationship between them (rho varied within −0.04–0.23 and *p* from 0.04 to 0.71).

### 2.4. Dynamics of Metabolites During Anoxia

We analyzed patterns of metabolite dynamics during anoxia. Appendix A present the results for roots and shoots, respectively. Metabolites were mapped according to strong correlations (r > 0.85). Metabolites were also clustered using hierarchical cluster analysis (HCA) with Spearman’s distance. The edges denote correlations: blue indicates negative, and red denotes positive. The dynamics patterns for the resulting clusters are illustrated in squares. Cluster membership is illustrated by the color outlines of nodes and graphs.

The metabolites formed two large groups in roots (Appendix A). The first of these consisted mainly of cluster 2 metabolites and included lipophilic compounds, phenolics and nitrogenous bases. These metabolites were characterized by an increase in the first day of anoxia followed by accumulation up to 72 h of treatment. Adjacent to them was a small sparse group of metabolites in cluster 6, which included fumarate, glycerate, Glu and Ser. The metabolites in this cluster showed a temporal decrease in the first few hours and then an elevation. They were also bordered by the small cluster 7, the metabolites in which were characterized by a rise in the level at 12 h of anoxia but decreased with a prolonged lack of oxygen. Metabolites in this cluster included several complex sugars and inositol phosphate. This was followed by the second large group of metabolites of cluster 5. They were characterized by a decrease in level with the development of an anoxic reaction. This cluster was dominated by complex sugars, including sucrose. There were also carboxylates, including intermediates of the Krebs cycle, and a certain number of amino acids. A distinction was the presence of amino acids and carboxylates in adjacent clusters 1 and 3. The patterns of these clusters were characterized by an elevated abundance at the beginning of anoxia followed by a decrease. The metabolites in cluster 4, located between the large groups, were shown to be up-regulated at the very beginning of anoxia, followed by a sharp depletion by 12 h with further growth.

In the case of shoots, the map was less homogeneous and connected, which may indicate a more complexly regulated metabolism (Appendix A). As in the case of roots, amino acids were concentrated in the clusters characterized by an increase in the levels at the beginning of anoxia, usually with a subsequent decrease (clusters 1 and 3). Cluster 1 also included citrate and oxalate. Complex sugars dominated in the large cluster 5, with a pattern demonstrating a decrease with ongoing anoxia. This cluster also included major steroids, Glu and Gln. Cluster 4 was located nearby, differing from 5 only by a slight elevation in the levels of metabolites at the very beginning of the stressor action; including, for example, β-Ala and allantoin. On the right was the large group of cluster 2. The accumulation of these metabolites was observed under anoxic conditions. Lipids, nitrogenous bases and carboxylates dominated here. This distinguishes shoots from roots, where carboxylates showed a trend of correlation with down-regulated complex sugars. Metabolites of cluster 6 (malate, malonate and phytol) were located nearby. The contents of these metabolites dropped by 12 h of anoxia and were then up-regulated.

### 2.5. Wheat Metabolic Profiles Under Reoxygenation Conditions

Wheat seedlings were transferred into a normoxic atmosphere for 1 and 24 h for reoxygenation after 6, 12, 24 and 72 h of anoxia application. Comparison of the metabolomes during reaeration using the principal component method showed that 1 h of reoxygenation had almost no effect on the metabolomes of either the roots or shoots, which clustered together with anoxic ones, especially after long periods of exposure (Figure 8). Reaeration for 24 h shifted the metabolome toward normal, especially after 6–12 h of anoxia, but a complete recovery did not occur, and 24 h post-anoxic metabolomes formed a separate cluster. After long-term anoxia (24–72 h), anoxic and post-anoxic metabolomes grouped together (Figure 8).

Figure 9 shows a heatmap of changes in the abundance of metabolites in wheat roots during reaeration relative to anoxic conditions. The anoxia-induced decrease in sugars generally continued upon reoxygenation. Nevertheless, some complex sugars accumulated under anoxic conditions were elevated during reaeration together with hexose phosphates. Levels of carboxylates and nitrogen-containing compounds generally decreased. It should also be noted that there were differences in metabolic reactions to reaeration after short (6–12 h) and long-term (24–72 h) anoxia. The abundance of pyruvate was down-regulated by post-anoxia after 6 h of oxygen deprivation and unchanged after longer exposition. The lactate level was decreased by reaeration after 6–12 h of anoxia but elevated after 72 h. Citrate and a number of amino acids (Asp, Glu, ornithine, oxoproline, Pro, Phe and Tyr) showed an increase after 6 h of anoxia but a decrease after longer anoxia. The abundance of glycerol, aconitate, succinate, malate, shikimate, Ala, Arg, GABA and Gly was down-regulated by reoxygenation after all periods of anoxia. Nitrogenous bases together with 2,3-butanediol showed a diphasic pattern, being up-regulated after 6 h of anoxia and depleted after 12–24 h. The highest elevation was shown for xanthine accompanied by adenosine, thymine and uridine after 6 h of anoxia. The contents of amines and polyamines decreased post-anoxia. Reaeration promoted further accumulation of fatty acids, acylglycerols and sterols, particularly after 6–12 h of anoxia (Figure 9, Appendix A).

MSEA showed that the main response of roots to reaeration was an increase in the level of metabolism of fatty acids and steroids (Figure 10). This reaction was expressed up to 72 h of anoxia. A distinctive feature of the effect of reoxygenation after 6 h of anoxia was an increase in nucleotide metabolism and a decrease in the metabolism of pentoses (including the pentose phosphate pathway) and ascorbate. The levels of intermediates of a wide range of pathways associated with amino acids, as well as the Krebs and glyoxylate cycles, were decreased by reaeration after prolonged anoxia.

The effects of reoxygenation were not detected after 24 and 72 h of anoxia in wheat shoots (Figure 11), and metabolomes clustered together (Figure 8). After 6–12 h of anoxia, a shift toward a normal metabolome, like that in the roots, was observed. The predictive components of the two OPLS-DA models accounted for 35 and 56% of the variance for 6 and 12 h of anoxia, respectively. The effect of post-anoxia on shoots was significantly different from that on roots. The main difference was the accumulation of all groups of sugars and derivatives (Figure 11, Appendix A). Only ascorbate and several oligosaccharides continued to be depleted upon reoxygenation. Carboxylates showed different response patterns. The abundance of aconitate, glycerate, malate, shikimate and quinate was down-regulated. The lactate level was decreased by reaeration after 6 h of anoxia and was elevated after 12 h. One can distinguish an accumulation of malonate. The anoxia-induced elevation in 3-hydroxypropionate continued during reaeration, while 4-hydroxybutyrate started to decrease. Changes in succinate levels in wheat shoots post-anoxia were insignificant (Appendix A). Glycerol and 2,3-butanediol levels were elevated. Amino acids and other nitrogen-containing compounds showed a decrease in abundance especially after 12 h of anoxia. Only a small number of lipophilic compounds showed an increase, mainly after 6 h of anoxia.

MSEA revealed that after 6 h of anoxia, the metabolism of fatty acids and steroids was stimulated, as in the roots; however, the effect was much weaker (Figure 10). Reoxygenation after 12 h of anoxia led to repression of amino acid-related pathways and nucleotide metabolism, which was reminiscent of the response of roots under prolonged anoxic conditions. Thus, the change in metabolic responses to reaeration in shoots occurred more quickly than in roots. Contrarily, starch and sugar metabolism were stimulated.

We compared the effects of anoxia and reoxygenation on the wheat metabolomes and potential similarities of metabolic responses in roots and shoots. It was revealed that effects during anoxia and post-anoxia were moderately similar in the roots with 6–24 h of oxygen deprivation (Appendix A). Spearman’s correlation coefficients between loadings from the corresponding OPLS-DA models were 0.32, 0.51 and 0.38 (*p* < 10^−9^) for 6, 12 and 24 h of oxygen deprivation and 24 h post-anoxia (positive values in the graph correspond to a higher level during anoxia [when comparing effects of normoxia and anoxia] and reaeration [when comparing effects of anoxia and reaeration]). This was consistent with the PCA plots (Figure 1 and Figure 8). There was no such relationship detected in wheat shoots (Appendix A). A comparison of the effects of reoxygenation on roots and shoots showed that they demonstrated slight similarity. Spearman’s correlation coefficients between OPLS-DA loadings were 0.29 (*p* = 0.008) and 0.23 (*p* = 0.043) for 6 and 12 h of anoxia followed by 24 h of reaeration (Appendix A).

## 3. Discussion

### 3.1. Metabolomes of Wheat Seedlings Under Short-Term Anoxic Conditions

The imposition of anoxia caused a significant alteration in metabolic profiles in wheat seedlings. The response of both root and shoot metabolomes to anoxia was two-stage (Figure 1). At the beginning of anoxic treatment, the responses of roots and shoots were rather similar and led to the depletion of the pool of sugars and sugar phosphates (Figure 2c,d), reflecting suppression of starch and sucrose metabolism (Figure 3b). This was accompanied by pyruvate and lactate accumulation, indicating stimulation of glycolysis, pyruvate metabolism (Figure 3b) and lactic fermentation. Unfortunately, the GC-MS metabolomics platform we used does not allow for detection of metabolites of the ethanolic fermentation pathway (acetaldehyde and ethanol). Nonetheless, our data obtained earlier with methods of conventional biochemistry in the same wheat cultivar and experimental setup showed high stimulation of ethanolic fermentation and accumulation of ethanol in roots up to 150–250 µmol/g FW after 24 h of anoxia (cited in [2]). MSEA revealed weak stimulation of the oxidative pentose phosphate pathway and the Krebs cycle (Figure 3b), although the abundance of metabolites of the latter such as aconitate and malate was decreased (Figure 2c,d). There was also an activation of amino acid metabolism (Figure 3b), predominantly associated with glycolysis (Gly, Phe, Tyr and Ala), Pro and GABA (Figure 2c,d). The obtained results showed that at the beginning of anoxic stress treatment, the metabolome of wheat, which is intolerant to oxygen deficiency, demonstrated a response rather similar to the typical long-term response of the majority of more tolerant plants, like Arabidopsis, rice and *Lotus japonicus* [17,18,20,24]. The acceleration of glycolysis supplies ATP synthesized in substrate phosphorylation, and activation of lactic and ethanolic fermentation ensures reoxidation of NADH required for glycolysis operation but leads to accumulation of toxic metabolites (acetaldehyde and ethanol) and cytosol acidification (lactate). Stimulation of amino acid metabolism provides nitrogen assimilation, synthesis of nitrogenous compounds, osmotic and pH adjustment, and the alternative regeneration of oxidized NAD(P)^+^, thus avoiding cytosolic acidosis and toxicity, as well as reduced carbon loss via fermentation [2,5,22,24]. Alterations in amino acid balance occurred during oxygen deprivation and are specific metabolic adaptations occurring in plants, in comparison to other organisms [25]. Pyruvate originated as a product of glycolysis and is used by the alanine aminotransferase/glutamate synthase cycle. Accumulation of Ala is considered as carbon storage instead of loss via ethanol fermentation [36].

#### 3.1.1. GABA Shunt

A very important component of the alternative pathways during hypoxia/anoxia is the GABA shunt, a bypass branch of the Krebs cycle [5,22,24,37], where 2-oxoglutarate is aminated to Glu and then Glu is decarboxylated to GABA, which takes place in the cytosol. Further on, GABA is converted to succinic semialdehyde in mitochondria. This reaction is coupled with the transamination of pyruvate to Ala, 2-oxoglutarate to Glu or glyoxylate to Gly [5,37,38]. Succinic semialdehyde is then oxidized to succinate in a reaction consuming NAD^+^. Severe anoxia would inhibit this reaction but stimulate metabolism of GABA into 4-hydroxybutyrate with the regeneration of NAD(P)^+^ [5]. It is commonly believed that GABA accumulation is involved in the regulation of pH, redox level and carbon/nitrogen balance. The predominantly metabolic role was demonstrated with the exogenous GABA effect via combined transcriptomics and metabolomics in *A. thaliana* seedlings [39].

According to our data, Glu was down-regulated, in contrast to GABA, succinate and 4-hydroxybutyrate (Figure 2c,d), in the wheat seedling metabolome even under short anoxic treatment. These findings suggest the activation of this pathway. Both GABA and 4-hydroxybutyrate improve plant adaptation not only to oxygen deprivation but also to other abiotic (salinity, drought, cold and heat) [38,40] and biotic (plant–pathogen and plant–symbiont interaction) stresses [41,42]. All spectra of tested stress factors cause cytosolic Ca^2+^ elevation. The link between Ca^2+^-signaling and GABA metabolism is supposed to depend on glutamate decarboxylase (GAD). The enzyme contains a calmodulin (CaM) moiety activated by Ca^2+^ [43]. Anoxia-induced cytosolic Ca^2+^ elevation is a widely studied process [44,45]. Nevertheless, GABA’s rapid increase up to a millimolar concentration in different tissues, cells and even compartments could be due to additional mechanisms of synthesis via polyamine degradation, and the reaction of proline with the hydroxyl radical results in the abstraction of hydrogen from the amine group and further spontaneous decarboxylation (recently reviewed in [46]). The GABA level could also be modulated through different mechanisms of its transport through cell and/or organelle membranes [46]. Recently, GABA has attracted special interest due to its dual role as a signaling molecule [46,47]. A putative GABA receptor identified in plants, ALMT1, belongs to the anion channel aluminum-activated malate transporter (ALMT) family [48]. GABA regulates membrane potential through this receptor coupled with Cl^−^ permeability. To date, few cases of such a signaling role have been confirmed, including stomata opening, pollen tube growth and root growth under Al^3+^-stress [47]. Details of signaling machinery for GABA are yet to be elucidated. It has even been hypothesized that GABA is suitable as a long-distance signaling molecule through the phloem [49]. Taken together, the important role of GABA during oxygen deficiency is obvious, but the multifaceted pathways of its implementation still need further investigation.

#### 3.1.2. Hydroxyl Carboxylic Acids

Furthermore, stimulation of butanoate and propanoate metabolism in roots and shoots at 3 h of anoxia was discovered in MSEA (Figure 3b). 3- and 4-hydroxybutyrate as well as 3-hydroxypropionate were also up-regulated in the anoxic wheat metabolome (Figure 2c,d). Little is known about the functional activity of these metabolites in plants. Thus, it is important to discuss available data on their occurrence, possible pathways of biosynthesis and interconversion, as well as potential toxic or protective roles under oxygen deprivation conditions in different taxa. Knowledge of mechanisms of action of specific metabolites in animals and microbes helps to distinguish their function in plants.

3-Hydroxybutyrate is one of the two main ketone bodies and a very important metabolite occurring in animals, plants and microbes. In animals, 3-hydroxybutyrate is formed as a product of fatty acid metabolism and acts as a regulatory molecule affecting gene expression, DNA methylation and histone deacetylation, thereby altering lipid metabolism, neuronal function and metabolic rate [50]. Furthermore, 3-hydroxybutyrate has diverse therapeutic properties in many human diseases and is reported to be synthesized in plants. Its conversion from acetoacetate or acetoacetyl-CoA is coupled with oxidation of NADH and therefore would be stimulated by an oxygen-depleted environment. Similarly to its role in animals, 3-hydroxybutyrate may act as a regulatory molecule related to DNA methylation/demethylation in plants [50]. On the other hand, accumulation of 3-hydroxybutyrate under anoxic conditions may be related to stimulation of lipid metabolism amid overconsumption and exhausting pools of carbohydrates.

3-Hydroxypropionate is an intermediate of Val and Ile catabolism in plant tissues. It can be converted into malonate semialdehyde by 3-hydroxyisobutyrate dehydrogenase AtHDH1 in *A. thaliana* [51]. In microbes, 3-hydroxypropionate may derive from pyruvate in the following pathway: pyruvate → α-Ala → β-Ala → 3-oxopropionate → 3-hydroxypropionate [52]. The last reaction consumes NADH. Along with 3-hydroxypropionate, α- and β-Ala were up-regulated in wheat seedling roots under short-term anoxic conditions (Figure 2c). The accumulation of 3-hydroxypropionate occurred not only in the roots but also in the shoots (during longer periods of anoxia, Figure 4 and Figure 6). This metabolite is also known to be synthesized from glycerol, glucose or fatty acids through acetyl-CoA and malonyl-CoA routes, reoxidizing NAD(P)H and FADH_2_ [53]. 3-Hydroxypropionate is an isomer of lactate (2-hydroxypropionate) and, together with other accumulated hydroxyl carboxylates, may participate in cytosol acidosis during anoxia.

Another anaerobic metabolite identified in wheat roots is 2-hydroxyglutarate (Figure 2c). In animals, it is synthesized in the reaction of Gln-derived 2-oxoglutarate reduction coupled with oxidation of NAD(P)H. 2-Hydroxyglutarate is a chiral molecule that exists as either *L*- or *D*-enantiomer. *L*-2-hydroxyglutarate production is induced by hypoxia in animal cells via promiscuous substrate use by lactate dehydrogenase and malate dehydrogenases 1 and 2 consuming NADH [54,55]. *D*-2-hydroxyglutarate is produced by mutant isocitrate dehydrogenase 1 and 2 oxidizing NADPH in a variety of human cancers [55]. Both enantiomers are toxic and increase the susceptibility of animal cells to developing tumors. In plants, *D*-2-hydroxyglutarate is believed to derive from Lys catabolism, and mitochondrial malate dehydrogenases 1 and 2 catalyze synthesis of *L*-2-hydroxyglutarate in a side reaction [56]. Stereospecific FAD-dependent dehydrogenases oxidize 2-hydroxyglutarate to 2-oxoglutarate in both plants and animals [55,56]. In plants, 2-hydroxyglutarate dehydrogenases transfer electrons to the functional electron-transfer flavoprotein/electron-transfer flavoprotein–ubiquinone oxidoreductase (ETF/ETFQO) complex linking amino acid catabolism to the mitochondrial electron transport chain during seed development and germination, carbohydrate starvation, senescence and water deficiency [56,57,58]. 2-Hydroxyglutarate is reported to have low toxicity to plant cells [59]. Thus, accumulation of 2-hydroxyglutarate in wheat seedling roots under short-term anoxic conditions could be a result of NAD(P)H oxidation stimulated by the reducing environment or amino acid utilization amid an exhausted sugar supply. Besides the accumulation of the listed compounds, the wheat root metabolome differed from the shoot one by its higher level and a greater number of amino, fatty and hydroxyl acids as well as lysolipids after 3 h of anoxic treatment (Figure 2c,d).

### 3.2. Metabolomes of Wheat Seedlings Under Prolonged Anoxic Conditions

#### 3.2.1. Carbohydrates

The continuation of anaerobic exposure from 3 to 6 h led to a more significant partition of normoxic and anoxic metabolomes and to initiation of the transition to the second stage of the metabolome’s dynamics mentioned above (Figure 1). The difference between anoxic root and shoot metabolomes became more evident (Appendix A). Further oxygen limitation (12, 24 and 72 h) resulted in serious alterations in metabolomes of wheat seedlings. We found relaxed and even intensified trends in sugar contents in wheat roots (Figure 4). Small percentage increases in pentoses and complex sugars were even observed to 24 h before finally decreasing gradually. This was assumed to be due to the intensive inhibition of different metabolic processes responsible for carbohydrate conversion such as starch and sucrose metabolism and glycolysis (Figure 5). Similarly, the depletion of tested sugars and sugar phosphates was reported earlier for wheat roots. The results of comparative analysis provided for five wheat genotypes after 1 day of anoxia revealed that two tested wheat cultivars showed the greatest differences in resistance to oxygen deprivation: Ducula (most sensitive) and Calingiri (most tolerant) [31]. With an additional temperature stress factor application (lower temperature: 15 °C instead of 28 °C), the intensity of sugar depletion was relatively low and, at the coldest temperature, the accumulation of sugar phosphates even increased in Caligrini roots. The mentioned effect probably reflects a common decrease in metabolism, which overlaps with acclimation to anoxia.

Surprisingly, the reaction in shoots under prolonged anoxic conditions was the opposite (Figure 6). Intensive depletion was clearly determined only for sugar phosphates and several complex sugars. Pentoses had the highest accumulation over time, while the dynamics of hexoses and complex sugars were mostly uneven. The revealed alterations were similar to those reported earlier. Long-term (up to 19 days) complete submergence was found to affect sugar metabolism in wheat shoots. Its decrement was assumed according to verified spectra of carbohydrates, which included starch, the sum of ethanol- and water-soluble carbohydrates such as glucose, fructose and sucrose. Nevertheless, the most intensive drop was determined within 2 days and then was stably maintained up to the end of the treatment. The reaction was observed to be similar in shoots of both the intolerant Frument and tolerant Jackson cultivars [30]. A similar effect was also determined in coleoptiles—a leaf modification at the juvenile stage of cereal development. One day of anoxia led to the down-regulation of sugar phosphates and sucrose [16]. The results of comparative metabolic analysis of coleoptiles of five wheat genotypes after 1 day of anoxia at different temperatures were slightly diverse. The experiment revealed no significant negative fluctuations in sugars and sugar phosphates. More precise analysis with Ducula (a more sensitive cv.) and Calingiri (a more tolerant cv.) showed a positive effect on fructose content in sensitive plants, especially at 28 °C [31]. After 12 days of waterlogging of wheat cv. Chinese Spring, plants exposed to 12 days of waterlogging during the tillering stage showed accumulation of glucose, fructose and sucrose, particularly in roots [60]. After 12 days, fructose content elevation was also determined in shoots in our analysis (Figure 6). This phenomenon is of interest because, in animals, fructose accumulation was discovered to maintain glycolysis and thus to support viability [61], and could be proposed as a mechanism of a tolerance for oxygen deficiency.

The analysis of sugar metabolism in wheat shoots revealed the inhibition of sucrose and starch metabolism, but this was less significant than in roots. Pathways such as glycolysis, the pentose phosphate pathway and galactose metabolism were slightly stimulated even under total anoxic conditions accompanied by the absence of photosynthesis (Figure 7). Thereby, the determined differences in root and shoot responsiveness might indicate a variation in the sensitivity of underground and aboveground plant organs. This phenomenon could reflect that those organs face oxygen deficiency, generally of different intensity. On the other hand, leaves are known to donate carbohydrates to roots; thus, the diversity of sugar contents in different organs would depend on phloem transport intensity. Oxygen limitation in castor bean (*Ricinus communis*) shoots caused a decrease in phloem sap flow and the conducting area, while the phloem sap sugar concentration was slightly increased [62]. Such an effect was determined even within minutes and was accompanied by accumulation of carbohydrates. Interestingly, the phloem transport reaction was found to quickly recover during the stress. Some indirect evidence has been published in the literature, but it concerns other plant species. Radioisotopic analysis confirmed the decrement in carbohydrate translocation in French bean (*Phaseolus vulgaris*) and maize [63,64]. Recently, a similar phloem transport alteration was mentioned for barrel medic (*Medicago truncatula*) with the GS-MS metabolomics approach [65]. All the listed species are known for their rather low tolerance to oxygen deprivation.

#### 3.2.2. Amino Acids and Nucleotides

Another group of metabolites generally involved in the adaptation of flooding-tolerant plant species is amino acids. In our experiments in wheat roots, a gradual decrease in the level of these substances was also discovered with the prolongation of anoxic treatment. The most pronounced effect was reached at 72 h (Figure 4). Among amino acids with the most intensive drop (after transient accumulation at 3 h of anoxia) were α- and β-Ala, Asn, GABA, His, oxoproline, Trp and Val. According to MSEA, this is a consequence of the limitation in amino acid metabolism—both synthesis and degradation (Figure 5). Simultaneous intensification of nucleotide metabolism also deserves attention. This could result from the limitation of energy metabolism and ATP synthesis, as well as catabolism of polynucleotides (RNA and DNA) and nitrogenous bases (accumulation of xanthine and urate, Figure 4). In animals and humans, hypoxia led to increased nucleotide degradation, which ends in xanthine and uric acid overproduction [66].

Literature data indicated that one day of anoxia triggered different reactions in roots of the wheat cultivars Ducula (more sensitive) and Calingiri (more tolerant) [31]. The ability to accumulate Gly, β-Ala, Pro and especially GABA (17.2-fold) coincided with higher resistance at 28 °C. In the colder regime (15 °C), even anoxia-sensitive Ducala showed accumulation of amino acids. Calingiri roots were characterized not only by intensification of the already mentioned amino acid elevation (23.5-fold for Pro and 36.3-fold for GABA), but also by wider spectra of accumulated metabolites of this group. The reason for such a cross-activation of stress tolerance in roots is not fully understood. The anoxia-induced dynamics in amino acid contents in wheat shoots was determined to be not as generalized as in roots (Figure 6). The contents of Gly, GABA and Lys exhibited different levels of accumulation during long-term anoxia. The Pro level was up-regulated only after 24–72 h of anoxia. According to the MSEA results, this coincided with a slight elevation in amino acid metabolism (Figure 7). Nevertheless, most amino acids showed a decrease in concentration (β-Ala, Asp, Asn, Glu, His, Met, oxoproline, Ser and Tyr). Previously obtained data indicated that, with complete submergence, 12 out of 17 tested amino acids (Asn, Gln, Ile, Leu, Lys, Met, Phe, Pro, Thr, Trp, Tyr and Val) were elevated in wheat shoots in comparison to others (Ala, Asp, Glu, Ser and Gly). This effect was determined in the first 12 days in tested wheat cultivars [30]. At the final stage of oxygen deprivation (14 and 16 days), the contents of amino acids in tolerant Jackson were found to exceed that in Frument. In another experiment, after 12 d of waterlogging, wheat roots accumulated the majority of measured amino acids (Ala, Glu, Gln, Gly, GABA and Ser), while in shoots the levels of these metabolites were down-regulated with the exception of Gln and Met [59]. Interestingly, Ala was most abundant in the xylem, and feeding of excised leaves derived from waterlogged wheat plants with Ala solution led to an increase in leaf glucose concentration. The authors consider this amino acid as a source of not only nitrogen, but also the carbon skeleton used for pyruvate recycling and further biosynthesis of glucose during hypoxia. Slight accumulation of GABA, Gly, Pro and Thr together with Asp decrease were observed in wheat coleoptiles after 24 h of anoxia [16]. Thus, along with the common trait of GABA accumulation, a different mechanism of resistance to anoxia is observed with varying levels of amino acids in roots and shoots. The elevation in the Pro level in shoots during prolonged anoxic treatment until 24–72 h might be a common trait of osmotic resistance in the event of carbohydrate losses, which was earlier suggested for rice [67], as well as a significant component of Pro antioxidant action, which is known to be important under stress conditions [68].

#### 3.2.3. Carboxylic Acids

Besides carbohydrates and amino acids, the decrement in the accumulation of carboxylates is shown for wheat roots (Figure 4). The maximal drop in the levels of citric and malic acids was reached at 72 h of anaerobic treatment. Nevertheless, a number of carboxylates, such as 3-hydroxypropionate and 4-hydroxybutyrate, had the opposite tendency. Accumulation of 4-hydroxybutyrate, which is closely linked to GABA and glutamate, was shown earlier in green tea and soybean seedlings in response to oxygen deficiency [69]. Previously published data illustrated that one day of anoxia caused a decrease in Krebs cycle metabolite levels, especially at 28 °C, with the maximum effect observed for Ducula (a more sensitive cultivar) [31]. The colder conditions (15 °C) increased the difference between cultivars. In this case, six tested acids (2-oxoglutarate, malate, succinate, fumarate, citrate and *cis*-aconitate) were accumulated in tolerant Calingiri in comparison to Ducula, which demonstrated a slight elevation in only succinic acid. Intermediates of the Krebs cycle such as citrates (isocitrate and citrate), succinate and 2-oxoglutarate were decreased or unchanged in leaves and accumulated in roots of wheat plants after 12 days of waterlogging [60]. Lactate was also detected only in roots. The opposite alteration in organic acid metabolism was determined in wheat shoots in our experiment with prolonged anoxia (Figure 6). With the exception of malic, malonic, oxalic and pyruvic acids, the revealed carboxylates were elevated. The most intensive increases were shown for lactate, fumarate, succinate, shikimate and 4-hydroxybutyrate. It is important that all mentioned acids were up-regulated, even at 6 h of oxygen deprivation, and were kept elevated until the end of the experiment (72 h). Krebs cycle metabolites were decreased in wheat coleoptiles after 1 day of anoxia in the more sensitive variety (Ducula), particularly at high temperature [31]. The exceptions were fumarate and especially succinate. At colder temperature (15 °C), all Krebs cycle acids showed down-regulation. The same 1-day anoxia in wheat coleoptiles revealed a shortage in citrate, isocitrate and malate levels, while aconitate, 2-oxoglutarate and fumarate were unchanged, and succinate was up-regulated [16].

Quinic and shikimic acids are among the metabolites with different dynamics in wheat roots and shoots under anoxic conditions. Roots were characterized by a slight depletion of the contents of both metabolites (Figure 4). However, quinate was slightly accumulated up to 12–24 h in shoots (Figure 6). The shikimate level increased substantially up to 72 h of anoxia. Quinic acid is a metabolite related to the shikimate pathway, and the synthesis of both acids consumes NADPH. Moreover, high levels of these acids have been determined in many plant tissues, especially in fruits [70]. An investigation of kiwi fruit development revealed that quinate was accumulated at early stages of development (less than 60 days after anthesis). This discovery might be important because fruits at early development stages are supposed to be an example of native hypoxic niches [71]. It was mentioned that quinate not only contributes to kiwi fruit flavor but also interferes with sugar/acid balance, which also could be involved in metabolic adaptation to oxygen deprivation. Moreover, in humans, the increase in the quinate level in pancreatic beta cells is triggered by transient cytosolic Ca^2+^ elevation due to intracellular resources, affecting the NAD(P)H/NAD(P)^+^ ratio and intensification of ATP synthase-dependent respiration [72].

#### 3.2.4. Lipids and Related Compounds

Among other hydroxyl carboxylic acids, 3-hydroxypropionate and 4-hydroxybutyrate also deserve attention. Both were accumulated during long-term anoxia in roots and shoots of wheat seedlings (Figure 4 and Figure 6). As mentioned above, synthesis of 3-hydroxypropionate may be associated with fatty acids [53]. Lipids certainly have a multifaceted role within plant metabolism and are involved in ensuring viability under stress conditions. They are also known to be crucial for membrane integrity and energy metabolism. In our experiments, the accumulation of hydroxyl carboxylates coincided well with the elevation in levels of a large number of fatty acids (including polyunsaturated), their derivatives and sterols, which were noticeable from 12 h of anoxia (Figure 4, Figure 5, Figure 6 and Figure 7). It was observed that fatty acid accumulation was much more intensive in roots, with the highest levels at 72 h of oxygen deprivation. The activation of several pathways of fatty acid and lipid metabolism in roots is clearly demonstrated in Figure 5 and Figure 7. It is necessary to note that both anabolic and catabolic pathways of lipid metabolism were stimulated (Figure 7).

Generally, severe oxygen deficiency down-regulates the contents of total lipids and phospholipids, hampers lipid biosynthesis and stimulates lipid degradation, leading to the accumulation of free fatty acids in plants [2]. The anoxia-induced lipid loss is more intense in hypoxia-intolerant plants, including wheat, in comparison to more tolerant Arabidopsis and rice [2,73]. Wetland species were shown to synthesize fatty acids and lipids [2,73] and utilize NADPH during these processes [74]. Moreover, the saturation of the fatty acid residues of phospholipids undergoes significant changes, as it is responsible for membrane fluidity and functions. Anoxia leads to a gradual decrease in the fraction of polyunsaturated fatty acid residues in lipids, while the fraction of saturated and monounsaturated fatty acid residues is up-regulated [2]. A similar phenomenon was shown in animal cells under chronic hypoxic conditions [74]. This process consumes NAD(P)H and participates in its recycling for glycolysis. In tolerant wetland species, saturation of membrane lipids occurs later under stronger oxygen limitation conditions than in intolerant ones, especially in mitochondria and plastids [2]. Moreover, the polyunsaturation of long-chain acyl-CoAs participates in hypoxia sensing by group VII ethylene response factors [73]. In our experiment, we observed the accumulation of 14 saturated, 2 monounsaturated and 5 polyunsaturated fatty acids and their derivatives in roots of wheat seedlings (Figure 4) and evident stimulation of the biosynthesis of polyunsaturated fatty acids, according to the MSEA results (Figure 5 and Figure 7). On the other hand, hypoxia was reported to stimulate synthesis of fatty acids and lipids in animal cells. Repetitive tissue hypoxia increased lipid biosynthesis and reversal of the Krebs cycle in muscle cells accompanied by suppression of succinate accumulation [75]. It should be noted that we observed stimulation of lipid metabolism and a decrease in succinate abundance in the roots of wheat seedlings during prolonged anoxia (Figure 4 and Figure 6). In neuronal cells—which are known to be sensitive to oxygen deficiency—a dramatic increase in fatty acid synthesis originates from Gln and Glu under hypoxic stress conditions [76]. It was assumed that this effect supports anaerobic metabolism under hypoxic conditions through oxidizing the reduced cofactors (NAD(P)H, FADH_2_). This maintains reduction potential and lowers lactoacidosis. Again, stimulation of fatty acid accumulation and down-regulation of Glu and Gln were found in both roots and shoots of wheat seedlings under anoxic conditions (Figure 4, Figure 5, Figure 6 and Figure 7). Surprisingly, membrane lipids of rat muscle cells were predominantly composed of saturated fatty acids under hypoxic conditions, while triacylglyceroles contained unsaturated fatty acids [74]. Recently, accumulation of triacylglycerol and an elevation in its polyunsaturation degree were revealed in roots of flooded tomato plants [77].

The accumulation of glycerol is a very important event for prolonged oxygen deprivation in wheat seedlings. This effect was determined in both roots and shoots, but was much more pronounced in shoots (Figure 4 and Figure 6). The maximum was reached at 24–72 h of anoxia. The pyruvate metabolism, Krebs cycle and oxidative phosphorylation were down-regulated under prolonged anoxic conditions, and the synthesis of glycerol from glycerate can take part in the reoxidation of NAD(P)H. Glycerol is also known as an osmolyte in *Dunaliella* species grown at high salinity [78] and a cryoprotectant in higher plants [79]. Additionally, glycerol is involved in lipid biosynthesis and, vice versa, may be released from lipid catabolism.

One more polyol accumulated in roots and shoots of wheat seedlings under anoxia and reoxygenation conditions was 2,3-butanediol, a volatile compound which was earlier shown to be produced by root-associated bacteria. Its biosynthesis from pyruvate involves condensation of pyruvate to acetoin and further reduction to 2,3-butanediol. The end step requires additional NADH [80]. Thereby, 2,3-butanediol synthesis was proposed to prevent acidification via changing an acidic molecule to a neutral compound. Moreover, additional NADH after glucose conversion to pyruvate is used for the transformation of acetoin to ethanol. Little is known about the possibility of 2,3-butanediol biosynthesis in plants. Nevertheless, some results showed that root tissues of several plant species are able to convert acetate to 2,3-butanediol [81], including *Brassica oleracea*, *Daucus carota*, *Pastinaca sativa* and *Raphnus sativus* but not *Brassica rapa*. The intensity of 2,3-butanediol synthesis was much lower and slower in comparison to microorganisms. Recently, leachates from needles of *Pinus armandii* were demonstrated to induce antifungal resistance in *Panax notoginseng* leaves [82]. 2,3-Butanediol was detected in leachates, and it was shown to activate the expression of genes responsible for systemic acquired resistance and directly affect plant immunity [83].

Besides the already mentioned substances, the levels of which were up-regulated during anoxia in roots but not in shoots, there were long-chain α,ω-dicarboxylic acids: azelaic and sebacic acids. These metabolites are synthesized in animals and human beings via ω-oxidation of fatty acids in the liver and kidney, and their biological role has not been explored yet. High production of long-chain dicarboxylic acids is associated with diabetes and several inborn metabolic and cardiovascular diseases [84]. Such diseases are characterized by the excretion in plasma of corresponding hydroxyl acids after ω-oxidation, including sebacic acid. An increase in the levels of these acids is a biomarker for diseases caused by mitochondrial β-oxidation disorder coupled with intoxication of mitochondria and energy deficiency [85]. Similar results can be observed during oxygen limitation in plant organisms. However, the role of azelaic and sebacic acids in wheat plant roots is yet to be examined. In plants, α,ω-dicarboxylates longer than azelaiate and sebaciate participate in suberin and cutin biosynthesis [86]. Two oxidoreductases—alcohol dehydrogenase (ADH) and BVMO (Baeyer–Villiger monooxygenase)—are known to be involved in the biocatalytic cascade of sebacic acid production in microorganisms [87]. For metabolic adaptation to oxygen deprivation it is important that the first enzyme requires NAD^+^, while the second depends on presence of NADPH. However, the dynamics of the activity of these enzymes and final involvement of the described multi-enzymatic cascade are far from certain.

Thus, accumulation of free fatty acids, 3-hydroxybutyrate, glycerol and long-chain α,ω-dicarboxylic acids more likely reflect the stimulation of lipid degradation, but stimulation of lipid biosynthesis in roots cannot be excluded.

#### 3.2.5. Main Directions of Metabolome Alterations

The temporal unfolding of the response to stress is associated with changes in the metabolome, which are manifested in coordinated changes in metabolite accumulation. Correlation analysis of metabolite accumulation is a method for determining functional relationships within the metabolome [88,89]. Metabolite mapping based on correlations in the dynamics of contents in root during anoxia (Appendix A) revealed two large groups. One of them consisted mainly of lipophilic compounds, the contents of which increased. Glycerol was also present among them, indicating an active coordinated process of membrane remodeling. The second group consisted of various compounds showing different patterns of accumulation decrease under anoxic conditions. It can be noted that sugars showed a decrease under anoxic conditions, while amino acids initially demonstrated an increase in accumulation. In the case of shoots (Appendix A), the structure was generally similar but the network was more heterogeneous, which is probably due to the more complex response of shoots to anoxia. The more consistent alterations in amino acids and sugars, which formed separate clusters, and the lack of tight coordination in the behavior of fatty acids are worth noting. It can be concluded that the response to stress at the metabolome level consisted of rearrangements of metabolite relationships with the formation of tightly bound bands. Coordination was associated with certain metabolic modules such as lipid metabolism, amino acid metabolism, sugar metabolism and carboxylate metabolism.

Thus, metabolomic analysis revealed a consistent rearrangement of metabolism in wheat plants under limited oxygen conditions. At the beginning of anoxic action, alterations in wheat root and shoot metabolomes showed similarity that lied in the accumulation of amino acids (Ala, GABA and Tyr), carboxylates (lactate and succinate), nucleotides and amines together with the decrease in sugars. The metabolic response to long-term anoxia varied significantly in roots and shoots of wheat seedlings, and was related to the redistribution of carbon flux from glycolysis; it was directed predominantly to lipids in the roots and to carboxylates in the shoots. This may have occurred during lipid biosynthesis in the roots during anoxia reoxidation of NAD(P)H and via anaplerotic pathways leading to carboxylates and GABA in the shoots. An important observation was the difference between the events occurring in roots and shoots, which were subject to the same anoxic stress and would seem to suffer equally. Despite the general decrement of almost all pathways of central metabolism in roots, the important exception was the intensification of nucleotide and lipid metabolism, including synthesis of fatty acids as well as their desaturation, elongation and degradation. The situation in shoots is less clear. Among all the groups of metabolites (carboxylic, amino and fatty acids, pentoses and nucleosides), a number of their representatives—particularly carboxylates—were elevated. Higher accumulation of carboxylates may be the reason for greater damage due to acidosis and lower viability of wheat shoots. Early on, we observed that 12 h of anoxia caused an 8-fold increase in electrolyte leakage from the shoots and only 4-fold from the roots of wheat seedlings [90]. A characteristic feature of the alteration in the metabolomes of both roots and shoots of wheat seedlings was the absence of α-Ala accumulation during prolonged anoxia, while the GABA level was up-regulated only in shoots. Moreover, under stress conditions in wheat plant organs, several metabolites with an unclear role associated with anoxic endogenous oxidation of NAD(P)H (e.g., hydroxyl carboxylates, α,ω-dicarboxylic acids, polyols) were distinguished and might be used as biomarkers for the processes developing in wheat plants sensitive to oxygen deprivation.

In contrast to wheat, the ability of submergence-tolerant plants to reroute glycolytic intermediates to alternate end products such as malate was first suggested in the “metabolic theory of flooding tolerance” by R.M.M. Crawford [91]. Although the major tenets of this hypothesis concerning possible inhibition of alcohol dehydrogenase activity and ethanol toxicity have been shown to be incorrect [4,74], much evidence has been discovered on the accumulation of different organic (shikimate and succinate) and amino acids (Ala and GABA) as well as glycerol in hypoxia-tolerant plants by methods of conventional target biochemistry [2,92,93]. Rerouting the glycolytic flow to these metabolites provides NAD(P)H reoxidation and, in the case of amino acids and polyols, addresses cytosolic acidosis during oxygen deprivation.

### 3.3. Wheat Metabolomes Under Reoxygenation Conditions

The period of oxygen deficiency is always followed by the restoration of aerobic conditions in the natural environment, and plants suffer from post-anoxic oxidative damage and progressive desiccation [6,33,34]. The reappearance of normoxia leads to the reestablishment of aerobic respiration that affects primary metabolism. According to fragmentary data available in the literature, the most significant metabolic changes involve the recovery of sugar levels and depletion of the levels of anaerobically induced metabolites, including lactate, succinate and amino acids; especially GABA, Ala and Gly (as reviewed in [25]). Unfortunately, the number of studies devoted to the metabolic adaptation to reoxygenation of different plant species is rather limited, and the knowledge about wheat metabolome recovery after oxygen deprivation is unclear.

In our experiments, 1 h of reoxygenation after 6, 12, 24 and 72 h of anoxia had no effect on root and shoot metabolomes of wheat seedlings (Figure 8). Reaeration for 24 h shifted the metabolome toward normoxic after 6–12 h of oxygen deprivation. Greater responsiveness after 6 h of anoxia could be due to the fact that, during this period, the transition to a later anaerobic response was not yet completed and a partial return of metabolism to the pre-stress state still was possible. Contrarily, after 24–72 h of anoxia, metabolomes following 24 h of reaeration were grouped with anoxic ones both in roots and in shoots (Figure 8), indicating much greater cellular damage. Interestingly, as with the metabolomes of wheat seedlings, the proteomes of rice seedlings after 24 h of anoxia clustered together with the post-anoxic (24 h) ones but not with the normoxic proteome, according to the PCA results [94]. The data of the metabolomic and proteomic analyses suggest that post-anoxia is not an independent stressor and should be considered as a continuation of the effects (or aftereffect) of anoxia.

Generally, the anoxia-induced alteration in root metabolome continued upon reoxygenation, which stimulated the metabolism of fatty acids, steroids and nucleotides (Figure 9 and Figure 10). Levels of organic acids (including succinate, lactate and other hydroxyl carboxylates), Ala and GABA proceeded to decrease. It is necessary to note that changes were the most pronounced after 6 h of anoxia. Accumulation of hexose phosphates and citrate indicated stimulation of respiration and up-regulation of amino acid abundance, corresponding to reactivation of the entire metabolism.

The effects of reoxygenation on wheat shoot metabolomes were slightly different. The abundance of all groups of sugars and malate was up-regulated, indicating starch and sugar metabolism stimulation (Figure 10 and Figure 11). Levels of carboxylates of the Krebs cycle (citrate, aconitate) as well as hydroxyl acids (lactate, glycerate, shikimate, quinate and 4-hydroxybutyrate), which were elevated during anoxia, became down-regulated upon onset of post-anoxia, showing changes in metabolism toward normoxic. As in roots, anoxia-stimulated metabolism of fatty acids and steroids was up-regulated during reaeration, but the effect was much weaker (Figure 10). 3-Hydroxypropionate, glycerol and 2,3-butanediol levels were elevated at reoxygenation after all tested anoxic treatments.

Effects of more prolonged anoxia (24–72 h for roots and 12 h for shoots) on wheat seedling metabolomes were less reversible: there was no decrease in lactate and amines (anaerobic metabolites) and less restoration of aerobic processes. One can note the depletion of ascorbate and suppression of ascorbate metabolism (Figure 9, Figure 10 and Figure 11), which corresponded to stimulation of post-anoxic oxidative stress. Earlier, we revealed accumulation of the oxidized form of ascorbate and a decrease in the level of total ascorbate accompanied by the inactivation of enzymes of the Foyer–Halliwell–Asada pathway, particularly in shoots of the same wheat cultivar seedlings [95]. Shoot metabolomes also differed by the elevation in malonate content (Figure 11). Accumulation of malonic acid might reflect malonic dialdehyde, an end product of lipid peroxidation. It is well known that lipid peroxidation is stimulated by oxygen deprivation and especially reoxygenation in plants, irrespective of their tolerance [6,33,96,97,98].

We did not find data for similar post-anoxic treatment of wheat plants in the literature. Thus, we decided to compare the reoxygenation effect with that in *Arabidopsis*, known as a species with medium tolerance to oxygen deprivation. The experimental design was quite similar to that presented in ours [99]. Sixteen-day-old seedlings were treated with total anoxia (100% nitrogen gas). The anoxia treatment (24 h) and further reoxygenation (0, 6, 12 and 24 h) were carried out in the dark to avoid oxygen generation during photosynthesis. The results of GC-MS analysis revealed different metabolic responses to reoxygenation. Several groups could be distinguished. The first group of metabolites—including lactate, Ala and GABA—were highly accumulated during anoxia; then, with oxygen availability, their levels intensively decreased. The second group—including Asp, Gln, Leu, Thr, Val and citrate—were reduced during hypoxia but increased during reoxygenation. The third group—including gluconate, guanine and adenosine—was characterized by intensive accumulation at longer durations of reoxygenation (12 and 24 h). The most representative group consisted of substances that were not substantially affected by oxygen restoration, including those that were either kept up-regulated (fructose, Pro, Gly and malonate) or down-regulated (Asn, Met, Ser, Cys and ornithine) during reoxygenation [100].

Similar data were obtained for post-anoxic metabolomes of *A. thaliana* seedlings after 6 h of reoxygenation preceded by 4 h of anoxia [100]. Levels of sucrose and fructose were temporally elevated in wild-type plants but were not affected by reaeration in defective *gdh1gdh2* mutants in glutamate dehydrogenase. Glucose content was down-regulated in both genotypes and pools of sugar phosphates were increased to normoxic levels in the wild-type plants, exceeding those in the mutants under post-anoxic conditions. Pyruvate was accumulated in both plants. A significant drop in levels of anaerobic metabolites (lactate, succinate, Ala and GABA) due to reoxygenation was observed in both plants, but amino acids remained above the control during post-anoxia. Contents of Glu and Asp depleted by anoxia were up-regulated in both genotypes during reaeration, with higher accumulation in the mutants. Anoxia-depressed levels of carboxylates of the Krebs cycle either were kept unchanged (fumarate) or increased by reoxygenation (citrate and malate), indicating the stimulation of the pathway. These alterations were the most notable in the wild-type genotype [100].

Taken together, these data indicate that metabolic rearrangement during reoxygenation is a complicated process and does not always fully return to the initial (pre-anoxic) level. The intensity of such a metabolic pathway might determine the ability of a plant to restore metabolism and thus survive stress. The alteration in metabolomes observed in wheat seedlings exposed to reoxygenation was similar to that observed in moderately tolerant Arabidopsis [99,100] and other plants [25]. Reaeration for 24 h after short-term anoxia (6 h) switched the metabolome toward normoxic, but complete recovery did not occur. Key groups of anaerobically down-regulated metabolites (e.g., sugars, sugar phosphates) were accumulated, while levels of anaerobic intermediates (lactate, hydroxyl carboxylates, dicarboxylates of the Krebs cycle and GABA) were depleted to varying degrees upon imposition of post-anoxia. Effects of more prolonged anoxia (24–72 h for roots and 12 h for shoots) on wheat seedling metabolomes were less reversible, corresponding to significant cellular damage. Subsequent reoxygenation had a slight effect on metabolomes, which were characterized by the maintained anoxia-stimulated metabolism of fatty acids, steroids and nucleotides, as well as by high levels of lactate and hydroxyl carboxylates.

## 4. Materials and Methods

### 4.1. Plant Material, Growing Conditions and Imposition of Anoxia and Post-Anoxia

Caryopses of spring wheat (*Triticum aestivum* L. cv. Leningradka, N.I. Vavilov All-Russian Institute of Plant Genetic Resources, St. Petersburg, Russia; agreement No. 120d/24 dated 14 May 2024) were surface-sterilized, germinated and grown hydroponically at an irradiance of 60 µmol·m^−2^·s ^−1^ with a photoperiod of 14/10 h at 23 °C as described earlier [90,101]. Groups of seedlings were fixed on the 7th day as the starting point prior to stress imposition. Other plants were placed in glasses containing 50 mL of Knop nutrient solution per 20 seedlings and separated into control (aerobic) and experimental (anaerobic) groups. The ambient temperature was 23 °C. Control seedlings were kept in darkness under normoxic conditions (21% O_2_). Gaseous nitrogen (less than 0.01% O_2_, Lentechgas, St. Petersburg, Russia) was fluxed through an anaerobic chamber for 45 min. Oxygen-free conditions were checked using an Anaerotest^®^ indicator (Merck, Darmstadt, Germany). Those chambers were hermetically closed and placed in the dark to prevent oxygen formation in the light due to photosynthesis. The duration of anoxic exposure lasted for 1, 3, 6, 12, 24 and 72 h. After 6, 12, 24 and 72 h of anoxia, group of the experimental plants were transferred into an aerobic atmosphere (post-anoxia) in darkness for 1 and 24 h of reoxygenation.

### 4.2. Sample Preparation for Metabolic Profiling

Mixed samples of entire roots and aboveground parts (shoots 0.5 cm above the caryopses) from 20 plants were analyzed. The tips of the leaf blades began to dry out due to 24 h of reoxygenation after 24–72 h of anoxia, and only alive basal parts of the leaves and stems close to caryopses were used for the reaeration study. Plant tissues (0.2 g) were placed in microtubes, frozen with liquid nitrogen and ground in a bead mill as described previously [102]. Methanol (1 mL) was used for metabolite extraction (1 h in a TS-100C thermoshaker [BioSan, Riga, Latvia] at 800 rpm, 4 °C). The samples were centrifuged for 10 min at 12,000× *g* and 4 °C, and the residue was washed twice (500 µL of methanol in a TS-100C thermoshaker), with further centrifugation of the sample each time (10 min at 12,000× *g*, 4 °C). The combined supernatants were dried in a Labconco CentriVap vacuum evaporator (Kansas City, MO, USA). The air in the microtubes was replaced with gaseous nitrogen, and the samples were placed in a freezer at −80 °C until analysis.

Before chromatography analysis, the dried material was dissolved in pyridine with the internal tricosane standard (nC23, tricosane, Sigma-Aldrich, St. Louis, MO, USA). The samples were then silylated with BSTFA/TMCS (bis(trimethylsilyl)-N,O-trifluoroacetamide/trimethylchlorosilane, 99:1, Sigma-Aldrich) at 90 °C for 20 min.

### 4.3. Gas Chromatography–Mass Spectrometry (GC-MS)

GC-MS analysis was performed using an Agilent 5860 gas chromatograph controlled with the MassHunter software (v. 10.1) (Agilent Technologies, Santa Clara, CA, USA) as described earlier [20]. An Rxi-5Sil (Restek Corporation, Bellefonte, PA, USA) MS capillary column (30 m, ID 0,25 mm, 0,50 µm) with constant flow of helium (carrier gas, 1 mL/min) was used. The research was carried out with chromatographic equipment of the Resource Center of St. Petersburg State University (“Development of molecular and cellular technologies”; SA 125022803066-3).

### 4.4. Interpretation of GC-MS Results

The PARADISe software (v. 6.0.1, Department of Food Science Faculty of Science, University of Copenhagen, Copenhagen, Denmark [103]) was used for the GC-MS analysis. The Golm Metabolome Database (GMD) library (Potsdam, Germany) was employed [104] for identification of the mass spectra, along with the library of the Laboratory of Analytical Phytochemistry of the V.L. Komarov Botanical Institute of the Russian Academy of Sciences (St. Petersburg, Russia; State Assignment no. 124020100140-7), and the AMDIS (v. 2.71) and NIST MS Search (v. 2.4) (National Institute of Standards and Technology (NIST), Gaithersburg, MD, USA) software was applied in combination with the NIST20 libraries. The identification of metabolites was performed according to the similarity of the mass spectra with the library ones, according to the retention indices (RIs). The RIs were acquired by plotting the calibration curve using saturated hydrocarbons. The contents of metabolites were normalized to the internal standard tricosane (normal hydrocarbon C_23_).

### 4.5. Statistical Analysis

The metabolomic data were analyzed using R 4.3.1 (“Beagle Scouts”) [105]. The data were normalized to the sample median. Outliers were detected and excluded according to the Dixon test in the outliers package [106]. Data were logarithmic and standardized. The imputation was performed using the KNN (k-nearest neighbors) method using the *impute* package (v. 1.80.0) [107]. Raw and preprocessed datasets included in Appendix A. *pcaMethods* (v. 1.98.0) was used for principal component analysis (PCA) [108]. OPLS-DA (discriminant analysis via orthogonal projections to latent structures) and OPLS were performed using the *ropls* package (v. 1.32.0). Factor loadings of the predictive components and VIP (variable importance in the projection) were used to assess the statistical relationship between the variables and the factor of interest [109]. Models were evaluated with R^2^Y and Q^2^Y (*p* < 0.01). To avoid bias due to group imbalance, weighted centering was used [110]. The ComplexHeatmap package (v. 2.22.0) was applied for plotting heatmaps [111]. The metabolite set enrichment analysis (MSEA) was performed by using the *fgsea* algorithm (v. 1.32.0) [112]. The sets of metabolites for biochemical pathways were downloaded from the Kyoto Encyclopedia of Genes and Genomes (KEGG) database (Japan) [113] via the KEGGREST package v. 1.46.0 [114] using *Triticum aestivum* as a reference organism. The Cytoscape software (v. 3.10.2) was applied for plotting graphs [115].

Metabolome analysis was performed with at least four biological replicates.

## 5. Conclusions

Wheat plants could not tolerate oxygen deprivation. The presented data clearly revealed the roles of metabolic alterations in wheat seedlings under anoxic conditions. The metabolic reaction to short-term anoxia (1–3 h) was similar to those known for other plants, including the accumulation of amino acids (Ala, Pro and GABA) and carboxylates (pyruvate, lactate and succinate) and a shortage of sugars and sugar phosphates (sucrose, glucose, fructose-6-phospate and glucose-6-phospate). It is necessary to note the elevation in the number of hydroxyl carboxylates (2-hydroxyglutarate, 3-hydroxypropionate and 3- and 4-hydroxybutyrate) (Figure 12). The shift in metabolic balance reflects a lack of energy and cytosol acidification. Continuation of anoxic treatment triggered different reactions in roots and shoots. The levels of most early accumulated metabolites—such as lactate, Ala, Pro and GABA—were decreased in roots, while pools of fatty acids and lipids (predominantly saturated) together with nucleotides were elevated. The aboveground parts, in contrast, continued to accumulate lactate, GABA, dicarboxylates of the Krebs cycle (succinate and fumarate) and hydroxyl carboxylates (3-hydroxypropionate, 4-hydroxybutyrate, shikimate and quinate) (Figure 12); some lipids, nucleotides and pentoses were also accumulated. Both roots and shoots contained high levels of polyols (glycerol and 2,3-butanediol) under anoxic conditions (Figure 12). Higher accumulation of carboxylates may be the reason for greater damage and lower viability of shoots. Such differences might indicate diverse sensitivity of seedling organs to oxygen deprivation according to developed metabolic activity. Further, post-anoxia also revealed the inequality of root and shoot metabolic responses. Reoxygenation after 6 h of anoxia triggered a shift in the metabolome toward normoxic. The majority of anaerobically down-regulated metabolites (e.g., sugars, sugar phosphates) were stimulated, while levels of anoxia-induced intermediates (lactate, hydroxyl carboxylates, dicarboxylates of the Krebs cycle and GABA) tended to be depleted. Reoxygenation after long-term anoxia induced few changes, and post-anoxic and anoxic metabolomes grouped together. Anoxia-induced stimulation of nucleotide and lipid metabolism proceeded, particularly in roots, while shoots responded by activating sugar metabolism. Lactate and hydroxyl carboxylates were abundant in tissues of wheat seedlings after prolonged anoxia followed by reaeration. The assumption is that shoots lose their ability to restore metabolism and thus lose viability faster than roots under oxygen deprivation conditions. Such a metabolic imbalance could be the reason for the high sensitivity of wheat plants to hypoxia.

An important finding of the presented investigation is the detection of the accumulation of hydroxyl carboxylates (2-hydroxyglutarate, 3-hydroxypropionate, 3- and 4-hydroxybutyrate, shikimate and quinate), as well as polyols (glycerol and 2,3-butanediol) in wheat seedlings at different durations of anoxia. This requires special attention and, together with the lack of Ala accumulation during long-term anoxia, these metabolites might be considered as markers of sensitivity to plant hypoxia. Biosynthesis of these compounds from intermediates of glycolysis and the Krebs cycle may be associated with anaplerotic pathways of anoxic endogenous reoxidation of NAD(P)H required for glycolysis operation, which can prevent the accumulation of toxic metabolites derived from conventional fermentation (acetaldehyde, ethanol and lactate). Our study identified new hydroxyl carboxylates, α,ω-dicarboxylates and polyols accumulated in the tissues of wheat seedlings under severe anoxic conditions, some of which are also characteristic of animal cells during hypoxia. Possible metabolic pathways synthesizing these metabolites and their potential toxic or protective role under oxygen deprivation conditions in plants must be elucidated.

## Figures and Tables

**Figure 1 ijms-26-11610-f001:**
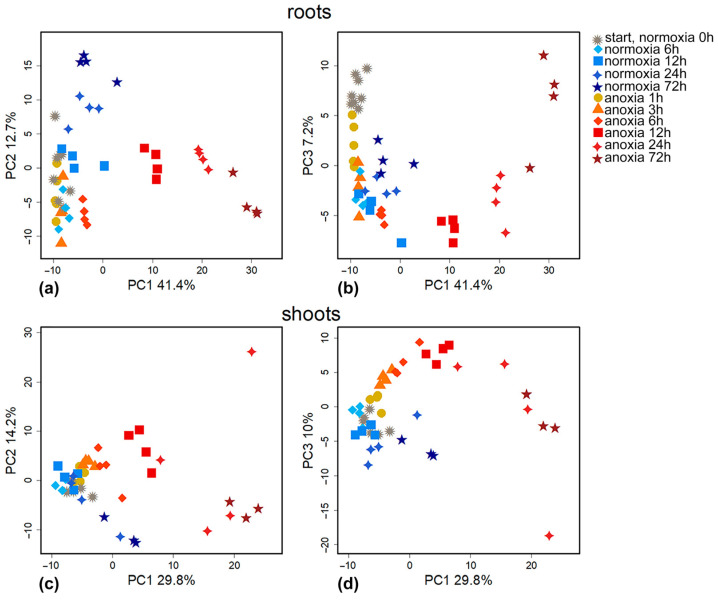
PCA score plots of the metabolite profiles of wheat seedling roots (**a**,**b**) and shoots (**c**,**d**) incubated under normoxic and anoxic conditions. “start”—plants immediately before incubation, 0 h; “h”—hours of incubation.

**Figure 2 ijms-26-11610-f002:**
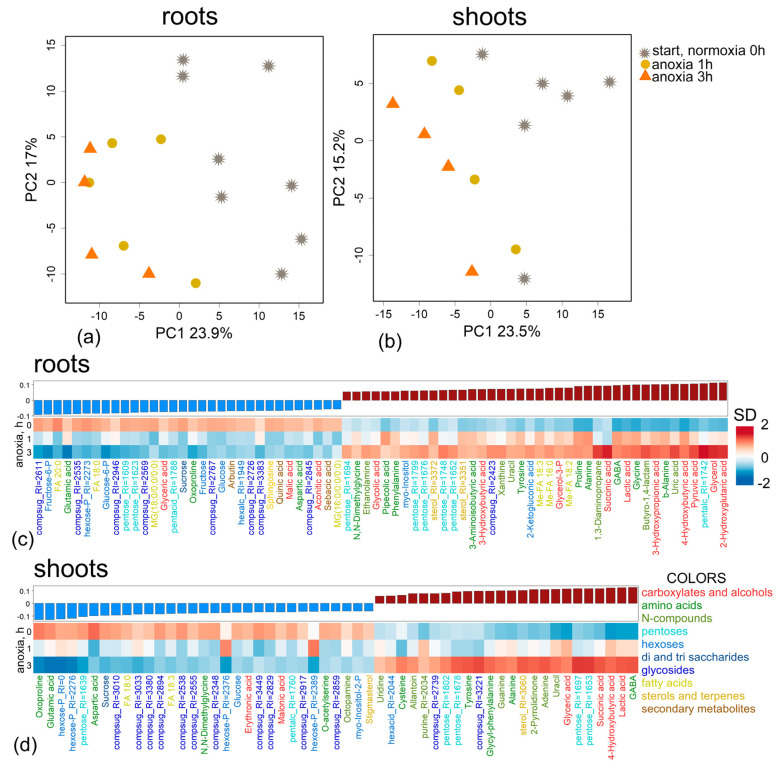
Effects of short-term anoxia (1–3 h) on the metabolite profiles of roots (**a**,**c**) and shoots (**b**,**d**) of wheat seedlings. (**a**,**b**)—PCA score plots. (**c**,**d**)—Bar plots of factor loadings of predictive components from OPLS models. Positive and negative loadings refer to levels that tend to increase and decrease with the duration of anoxia, respectively. The heatmap represents means of the normalized per sample median, logarithmic and standardized contents. Colors in data labels mark chemical classes.

**Figure 3 ijms-26-11610-f003:**
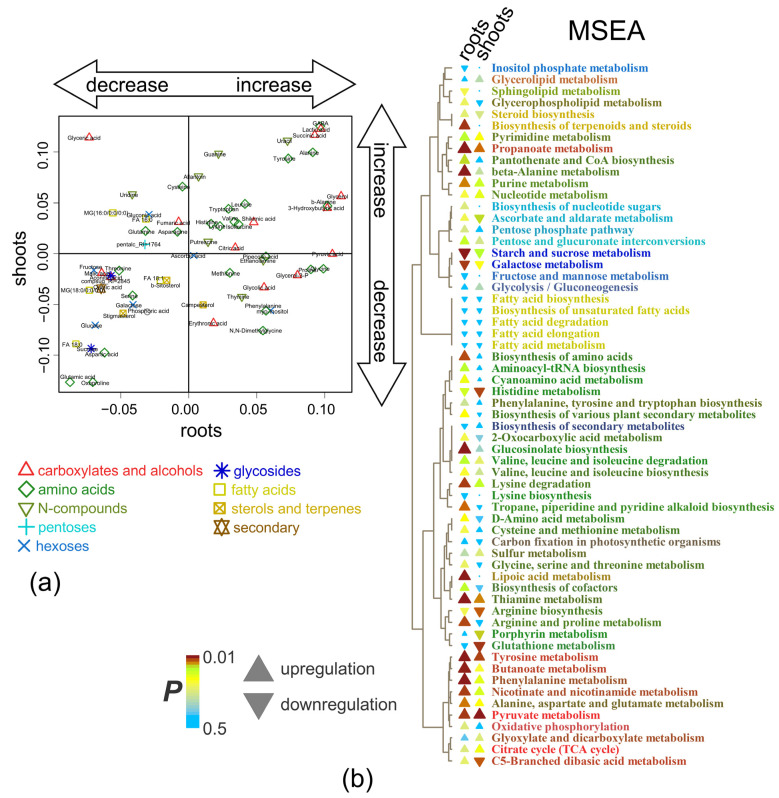
(**a**)—Comparison of root and shoot metabolite dynamics under short-term anoxic conditions (1–3 h). SUS (shared and unique structure) plot in the space of the loadings from OPLS models for roots and shoot at abscises and ordinates, respectively. Positive and negative loadings refer to levels that tend to increase and decrease with the duration of anoxia, respectively. (**b**)—MSEA results. Pathways are clustered according to the number of common metabolites in the profile. Triangles—direction of effect; size—strength; color—*p*-value. Pathways are colored according to the chemical classes of metabolites.

**Figure 4 ijms-26-11610-f004:**
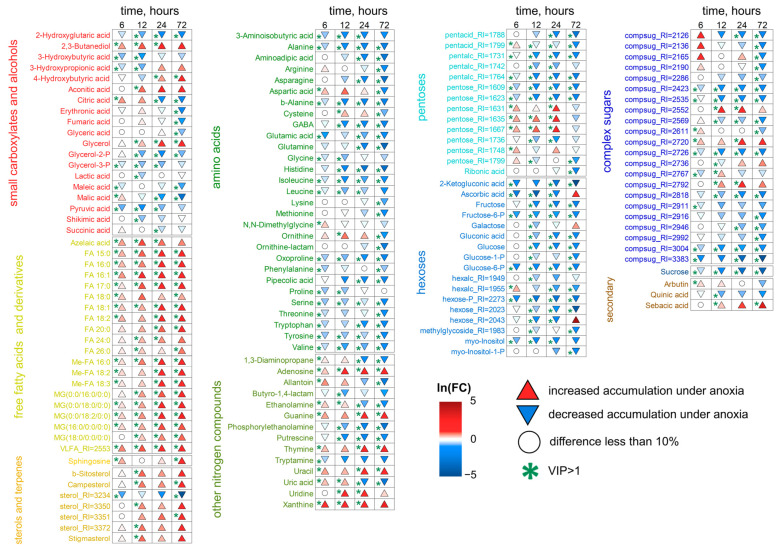
Effects of long-term anoxia (6–72 h, numbers above) on the metabolite profiles of roots of wheat seedlings. Heatmaps showing the Fold Change (FC) differences between root metabolomes of control (normoxic) and anoxic plants (FC = ln(C_anoxia_/C_normoxia_)). Asterisks indicate cases with VIP > 1.

**Figure 5 ijms-26-11610-f005:**
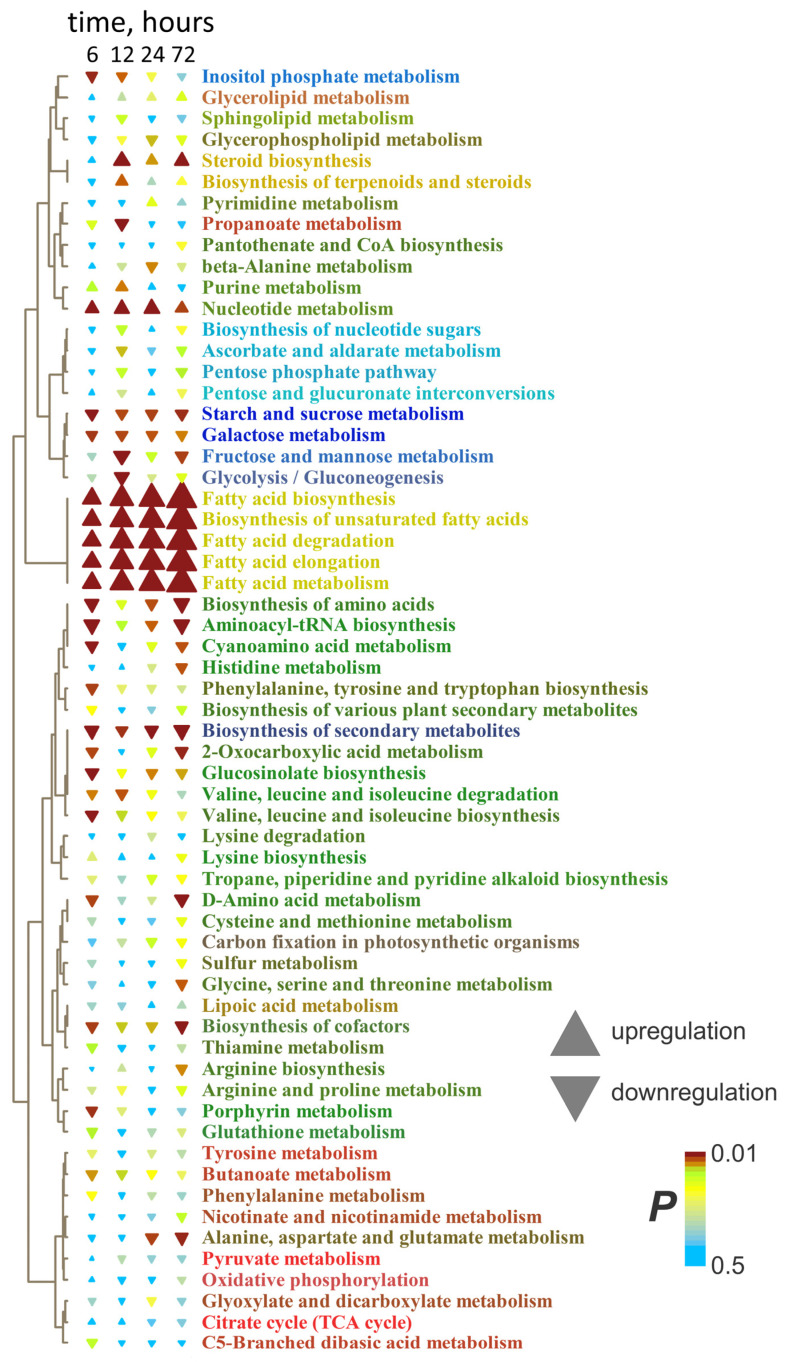
Metabolite set enrichment analysis (MSEA) of roots of wheat seedlings under long-term anoxic conditions (6–72 h, numbers above). Pathways are clustered according to common metabolites in the profile. Triangles—direction of effect; size—strength; color—*p*-value. Pathways are colored according to the chemical classes of metabolites.

**Figure 6 ijms-26-11610-f006:**
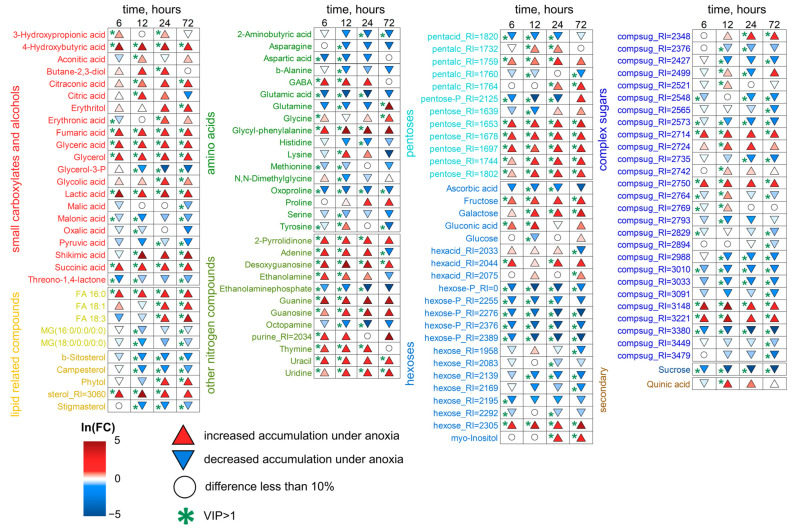
Effects of long-term anoxia (6–72 h, numbers above) on the metabolite profiles of shoots of wheat seedlings. Heatmaps showing the Fold Change (FC) differences between shoot metabolomes of control (normoxic) and anoxic plants (FC = ln(C_anoxia_/C_normoxia_)). Asterisks indicate cases with VIP > 1.

**Figure 7 ijms-26-11610-f007:**
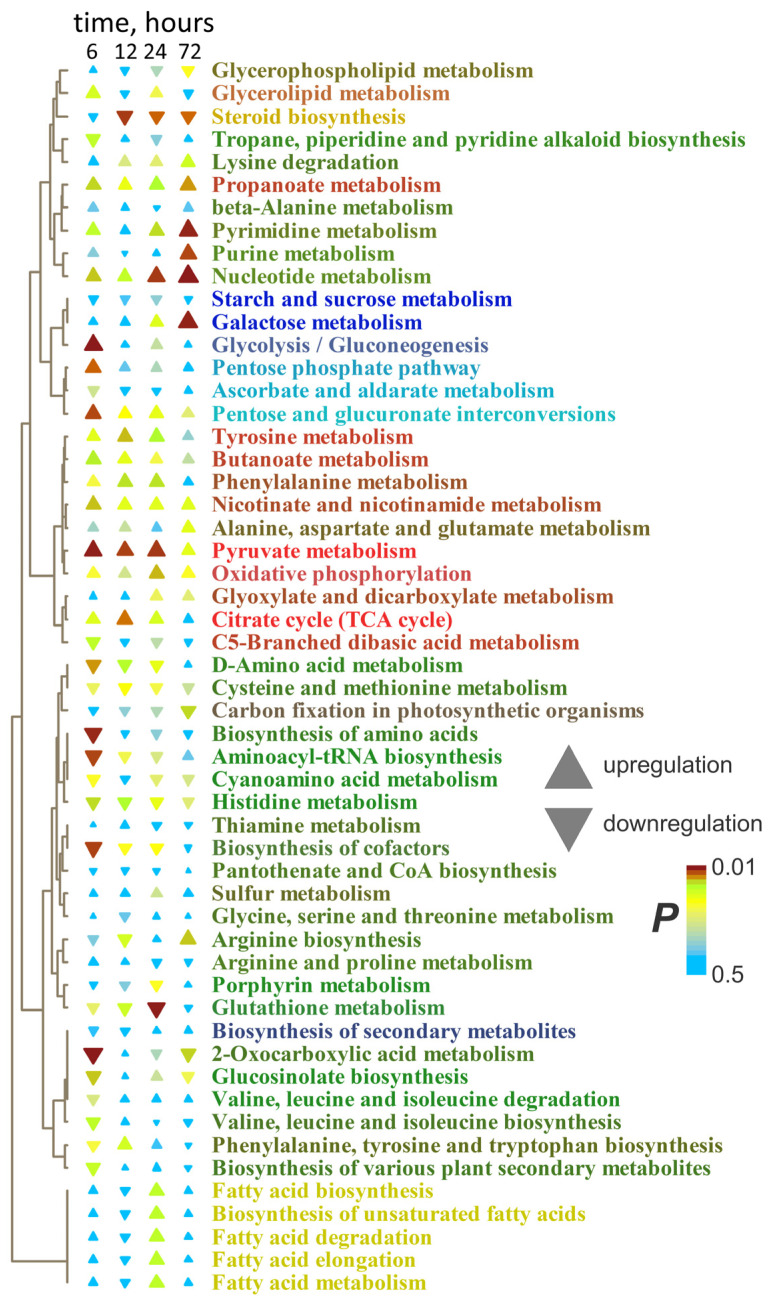
Metabolite set enrichment analysis (MSEA) of shoots of wheat seedlings under long-term anoxic conditions (6–72 h, numbers above). Pathways are clustered according to common metabolites in the profile. Triangles—direction of effect; size—strength; color—*p*-value. Pathways are colored according to the chemical classes of metabolites.

**Figure 8 ijms-26-11610-f008:**
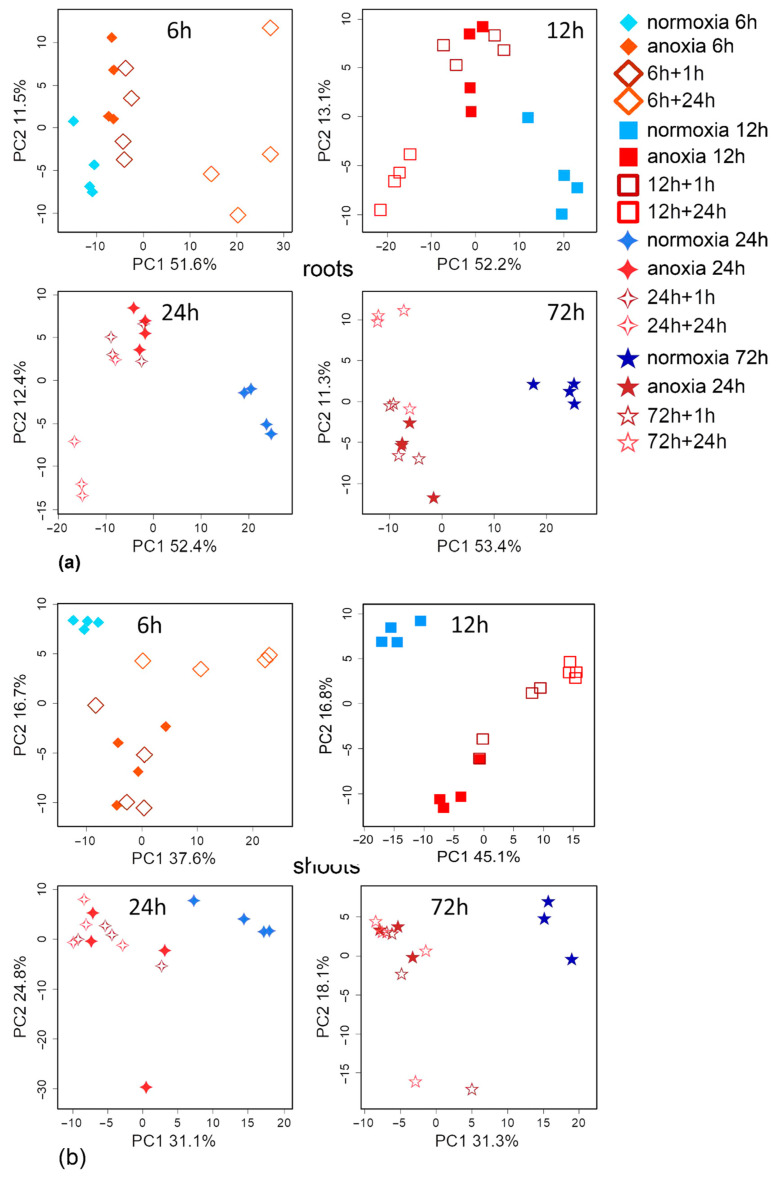
Unsupervised analysis of effects of reoxygenation (+1 and +24 h) after 6, 12, 24 and 72 h of anoxia on the metabolite profiles of roots (**a**) and shoots (**b**) of wheat seedlings. PCA score plots. The points correspond to the metabolite profiles under normoxia, anoxia and reoxygenation conditions. The numbers “XX + YY” on the graphs are the duration of anoxia (XX) and subsequent reoxygenation (YY).

**Figure 9 ijms-26-11610-f009:**
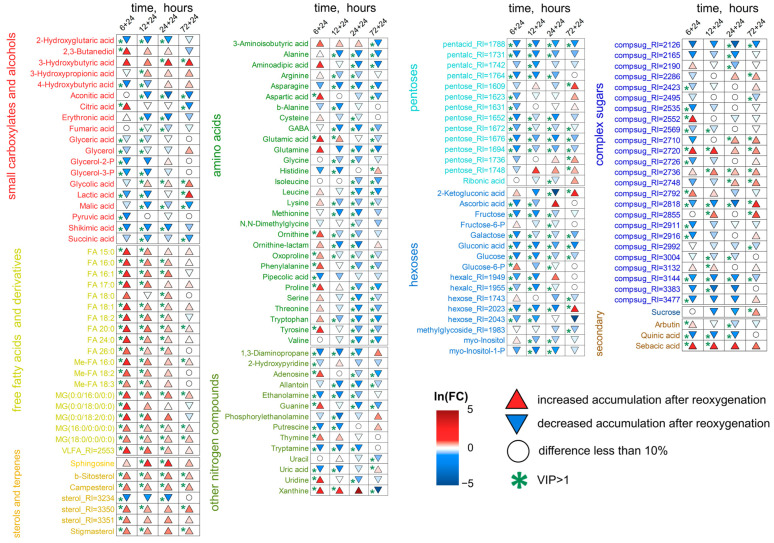
Effects of 24 h of reoxygenation after 6–72 h of anoxia on the metabolite profiles of roots of wheat seedlings. Heatmaps showing the Fold Change (FC) differences between root metabolomes of anoxic and post-anoxic plants (FC = ln(C_reoxygenation_/C_anoxia_)). Asterisks indicate cases with VIP > 1. The numbers “XX + YY” are the duration of anoxia (XX) and subsequent reoxygenation (YY).

**Figure 10 ijms-26-11610-f010:**
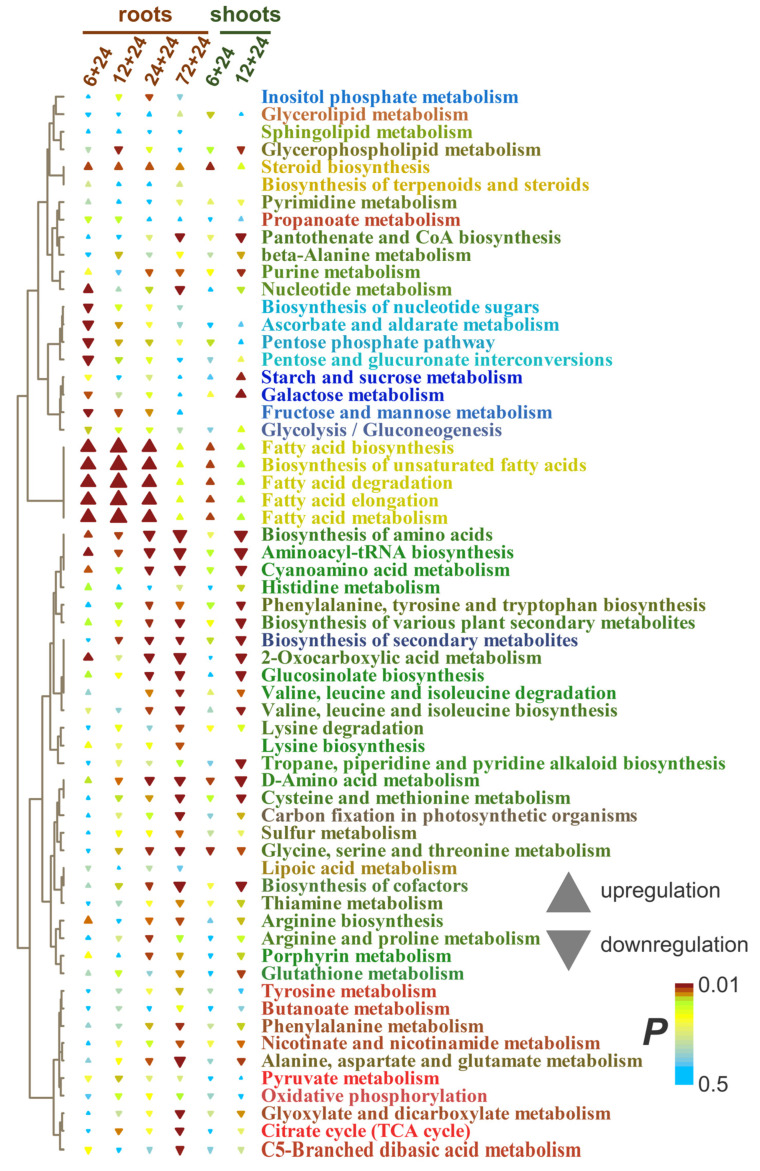
Metabolite set enrichment analysis (MSEA) of roots and shoots of wheat seedlings following 24 h of reoxygenation after 6–72 h of anoxia. Pathways are clustered according to common metabolites in the profile. Triangles—direction of effect; size—strength; color—*p*-value. Pathways are colored according to the chemical classes of metabolites they consist of. The numbers “XX + YY” are the duration of anoxia (XX) and subsequent reoxygenation (YY).

**Figure 11 ijms-26-11610-f011:**
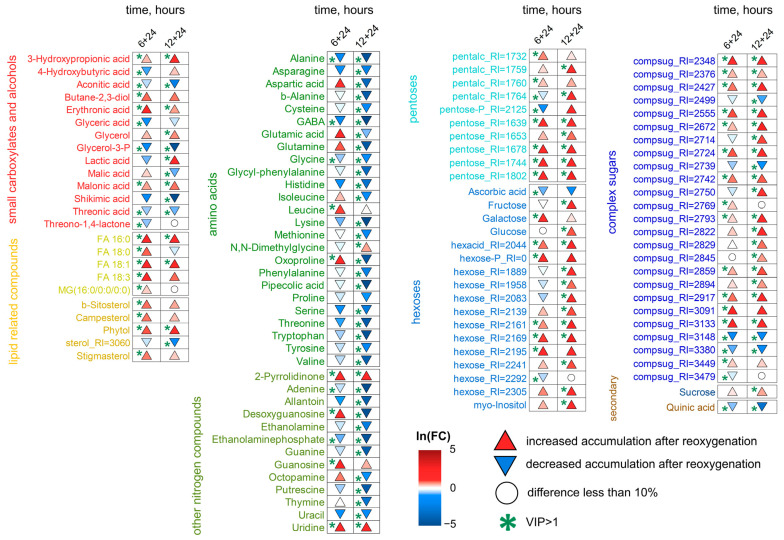
Effects of 24 h of reoxygenation after 6–72 h of anoxia on the metabolite profiles of shoots of wheat seedlings. Heatmaps showing the Fold Change (FC) differences between root metabolomes of anoxic and post-anoxic plants (FC = ln(C_reoxygenation_/C_anoxia_)). Asterisks indicate cases with VIP > 1. The numbers “XX + YY” are the duration of anoxia (XX) and subsequent reoxygenation (YY).

**Figure 12 ijms-26-11610-f012:**
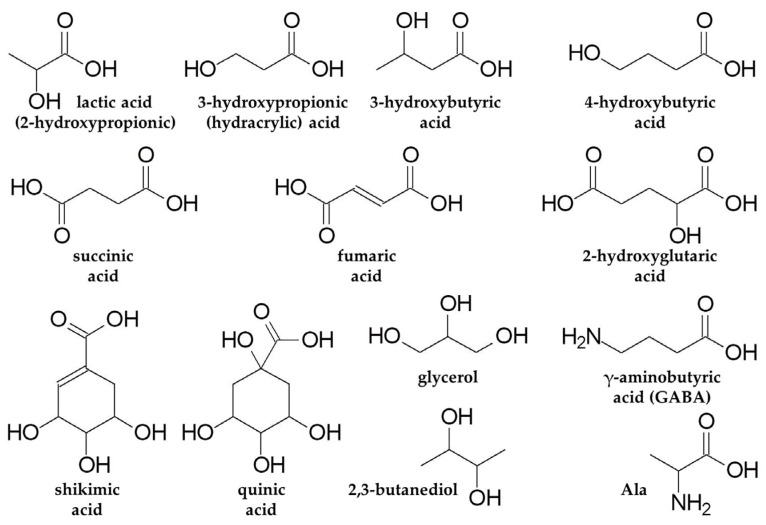
Formulas of amino and carboxylic acids and polyols accumulated in wheat seedlings under anoxic conditions.

## Data Availability

The original contributions presented in this study are included in the article/Appendix A. Further inquiries can be directed to the corresponding author.

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
