# Peer review of "Metabolic Profiling of Wheat Seedlings Under Oxygen Deficiency and Subsequent Reaeration Conditions"

_ijms, 2025, doi:10.3390/ijms262311610_

Round 1
Reviewer 1 Report
Comments and Suggestions for Authors
There are some suggested revisions as follows:
- Line 28-32
Please specify in which part (roots and/or stems) these results and phenomena were observed.
- Line 48
Does Oryza sativa specifically refer to the Latin name of "deepwater rice"? It is suggested to modify it as: "deepwater rice (a specific ecotype of Oryza sativa)".
- Line 3.1. Metabolomes of Wheat Seedlings under Anoxia
This section includes nearly 30 paragraphs, which may not be very reader-friendly. Please streamline and merge paragraphs.
- Line 566-570, 591-598, 798-800
The author describes the roles of specific metabolites in animal life activities in these places, which helps readers deepen their understanding of the effects of these metabolites. However, does this expansion deviate from the theme of this study (focusing on plants as the research object) and thus distract readers' interest and attention? I have reservations about this expansion and suggest that the author delete these sentences.
- Line 1106
Are these metabolites the common response substances found in both roots and shoots? It is suggested to indicate in the figure which metabolites are in the roots and which are in the shoots.
- Line 1013
The author specified the light conditions (darkness) for plant growth during oxygen deficiency. Please supplement the temperature conditions during this process.
- Line 1018
Please indicate the specific location of sampling, such as the root tip or mature zone of the root, or axial root, or fibrous root, or the entire root mixed; the base or tip of the stem, or the entire stem mixed.
- Line 1120-1124
In the "conclusions" section, the author should summarize the conclusions drawn from this study, rather than engaging in discussions (and citing references from others). Please reorganize the conclusions section and make it more concise.
Author Response
Dear Reviewer!
We appreciate honorable reviewer for the positive attitude to our results and given comments. We accepted the most part of them and added proper information in revised text. Corrections of the text are high lightened in green in the pdf-file. But we hope that the reviewer will agree with a necessity to keep description of functional importance of specific metabolites in organisms other than plants.
Comments and responses:
Comments 1: Line 28-32. Please specify in which part (roots and/or stems) these results and phenomena were observed.
Response 1:
These results were observed in both, roots and shoots. But reversion of metabolome towards normoxic one after short-term anoxia was more pronounced in root, while damage and irreversion were found particularly in shoots. The last statement is specified (lines 32, 35). Unfortunately, we are limited by the requirement of 200 words volume of abstract.
Comments 2: Line 48. Does Oryza sativa specifically refer to the Latin name of "deepwater rice"? It is suggested to modify it as: "deepwater rice (a specific ecotype of Oryza sativa)".
Response 2:
Corrected
Comments 3: Line 3.1. Metabolomes of Wheat Seedlings under Anoxia. This section includes nearly 30 paragraphs, which may not be very reader-friendly. Please streamline and merge paragraphs.
Response 3:
We rearranged material of this subsection, divided it into two parts, merged several paragraphs, and introduced numbering of subtitles.
Comments 4: Line 566-570, 591-598, 798-800. The author describes the roles of specific metabolites in animal life activities in these places, which helps readers deepen their understanding of the effects of these metabolites. However, does this expansion deviate from the theme of this study (focusing on plants as the research object) and thus distract readers' interest and attention? I have reservations about this expansion and suggest that the author delete these sentences.
Response 4:
We believe that those compounds are of a special interest but its role is undiscovered for plants yet. Recently some data was presented about the functional activity of those metabolites in other organisms. Thus we found it to be important to discuss possible common mechanisms of damage and adaptation in different taxa of life under oxygen deprivation. Knowing of mechanisms of action of specific metabolites in animals and microbes helps to distinguish their function in plants.
Comments 5: Line 1106. Are these metabolites the common response substances found in both roots and shoots? It is suggested to indicate in the figure which metabolites are in the roots and which are in the shoots.
Response 5:
We are thankful to honorable reviewer for the suggestion. In Discussion we already form sections specially focused on specificity of such a differences between roots and shoots. Those are based on data in Fig. 3, 4, 6 and Fig. S1. But majority of compounds presented at Figure 12 are accumulated in both, roots and shoots throughout the anoxic treatment either at early or at long-term one.
Comments 6: Line 1013. The author specified the light conditions (darkness) for plant growth during oxygen deficiency. Please supplement the temperature conditions during this process.
Response 6:
Proper information is added in Material and Methods (line 1044).
Comments 7: Line 1018. Please indicate the specific location of sampling, such as the root tip or mature zone of the root, or axial root, or fibrous root, or the entire root mixed; the base or tip of the stem, or the entire stem mixed.
Response 7:
Proper information is added in Material and Methods (lines 1054-1057).
Comments 8: Line 1120-1124. In the "conclusions" section, the author should summarize the conclusions drawn from this study, rather than engaging in discussions (and citing references from others). Please reorganize the conclusions section and make it more concise.
Response 8:
We provided proper changes in “Conclusion”, excluded references and moved the debatable part into “Discussion”.
Sincerely yours,
Authors
Reviewer 2 Report
Comments and Suggestions for Authors
The manuscript offers valuable metabolomic data and insights into the response of wheat to oxygen deprivation. Addressing the following points, particularly those related to data presentation, interpretation of metabolic roles, and broader implications, will substantially enhance the paper.
- The study conductsa comprehensive metabolomic analysis of wheat seedlings under anoxia and reoxygenation, emphasizing tissue - specific responses (roots versus shoots) and temporal dynamics. The identification of relatively under - investigated metabolites such as hydroxyl carboxylates, α,ω - dicarboxylic acids, and polyols as potential markers of hypoxia sensitivity represents a significant contribution. Nevertheless, the novelty could be augmented through a more direct comparison with hypoxia - tolerant species or cultivars to more effectively contextualize the observed metabolic shifts as indicators of sensitivity.
- The experimental design is robust, featuringmultiple time points for anoxia and reoxygenation, and the utilizationof GC - MS for metabolomic profiling is appropriate. However, were biological replicates clearly defined and randomized? How was the influence of darkness (employed in both the control and treatment groups) considered in the interpretation of metabolic changes? The inability to detect ethanol/acetaldehyde due to GC - MS limitations should be explicitly discussed as a constraint.
- The application of PCA, OPLS-DA, MSEA, and correlation - based clustering is thorough and supports the conclusions. However, the heatmaps (e.g., Figures 4, 6, 9, 11) are dense and challenging to interpret. A curated selection of key metabolites or pathways would enhance clarity.
- The discussion effectively associatesmetabolite changes with known pathways (e.g., GABA shunt, lipid metabolism, Krebs cycle). However, the functional implications of the increased lipid and nucleotide metabolism in roots under long - term anoxia remain speculative. Are these adaptive or detrimental responses? The role of 2,3 - butanediol and its potential microbial origin should be further explored, including whether its accumulation is derived from the plant or the microbiome.
- A more in - depthdiscussion of the reasons forsuch divergence and its physiological consequences is required. Whether these differences are attributable to inherent tissue metabolism or varying degrees of anoxia exposure needs to be addressed.
- The discussion could be strengthened by linking these metabolic irreversibilities to physiological outcomes (e.g., oxidative damage, cell death).Are there any metabolic markers that can predict recovery potential?
- The manuscript is generally well-written, yet some sections are overly complex and would benefit from simplification. For example, the description of metabolite dynamics in Sections 2.3 and 2.5 could be more concise. Although abbreviations (e.g., OPLS - DA, MSEA) are well - defined, their repeated use can impede readability.
- The conclusion summarizes the main findings but could more effectivelyemphasize the broader relevance of the study. For instance, howcan these metabolic markers be utilized in breeding or biotechnological approaches to enhance hypoxia tolerance in wheat? Are the identified metabolites conserved across other cereals or sensitive species?
Author Response
Dear Reviewer!
We are thankful for your deep consideration of our data and for interesting comments. We added all the necessary information to revised manuscript. The Figures, heatmaps and legends were redesigned. Corrections of the text are high lightened in green in the pdf-file. We also prepared detailed responses which we hope could be accepted.
Comments and responses:
Comments 1: The study conductsa comprehensive metabolomic analysis of wheat seedlings under anoxia and reoxygenation, emphasizing tissue - specific responses (roots versus shoots) and temporal dynamics. The identification of relatively under - investigated metabolites such as hydroxyl carboxylates, α,ω - dicarboxylic acids, and polyols as potential markers of hypoxia sensitivity represents a significant contribution. Nevertheless, the novelty could be augmented through a more direct comparison with hypoxia - tolerant species or cultivars to more effectively contextualize the observed metabolic shifts as indicators of sensitivity.
Response 1:
We appreciate the honorable reviewer for such a suggestion. We already provided similar investigation with rice seedling and now this data is under preparation for submission. But for this manuscript we believe it would be too extensive to be added. Literature data on metabolomics of hypoxia-tolerant plant was recently reviewed [24, 25]. This information is mentioned in Introduction.
Comments 2: The experimental design is robust, featuring multiple time points for anoxia and reoxygenation, and the utilization of GC - MS for metabolomic profiling is appropriate. However, were biological replicates clearly defined and randomized? How was the influence of darkness (employed in both the control and treatment groups) considered in the interpretation of metabolic changes? The inability to detect ethanol/acetaldehyde due to GC - MS limitations should be explicitly discussed as a constraint.
Response 2:
We used randomized mixture of entire roots or shoots from 20 seedlings in 4-8 biological repetitions. Darkness did affected all aspects of plant life, but when we started investigation of plant adaptation to oxygen deficiency, we found that anoxia at light damaged shoot tissues more significantly than anoxia at dark due to harmful effects on photosynthetic apparatus and overproduction of reactive oxygen species, particularly in intolerant plants like wheat [Chirkova T.V., Walter G., Leffer S., Novitskaya L.O. Chloroplasts and mitochondria in the leaves of wheat and rice seedlings exposed to anoxia and long-term darkness: some characteristics of organelle state // Russian Journal of Plant Physiology. 1995. Vol. 42, No. 3. P. 321-329.]. Moreover, effects of darkness under normoxia on different aspects of are almost negligible in comparison with effects of anoxia. Ethanol and acetaldehyde would be evaporated during sample preparation for GC-MS metabolomics protocol. For these compounds special procedure for volatiles extraction has to be used.
Comments 3: The application of PCA, OPLS-DA, MSEA, and correlation - based clustering is thorough and supports the conclusions. However, the heatmaps (e.g., Figures 4, 6, 9, 11) are dense and challenging to interpret. A curated selection of key metabolites or pathways would enhance clarity.
Response 3:
The Figures were redesigned, heatmaps altered as large as possible, and stars were colored in bright green for easier visibility. A legend was added, and connection classes named and colored as well.
Comments 4: The discussion effectively associates metabolite changes with known pathways (e.g., GABA shunt, lipid metabolism, Krebs cycle). However, the functional implications of the increased lipid and nucleotide metabolism in roots under long - term anoxia remain speculative. Are these adaptive or detrimental responses? The role of 2,3 - butanediol and its potential microbial origin should be further explored, including whether its accumulation is derived from the plant or the microbiome.
Response 4:
The metabolism is consists of both, anabolism and catabolism. Increased lipid and nucleotide metabolism revealed by MSEA means activation of their degradation too. We mentioned majority of known information about lipids in the text, that more likely elevation of free fatty acid abundance came from lipid catabolism (lines 797-801), but different cases of lipid biosynthesis which may provide paths to reoxidize NAD(P)H are listed in manuscript text too (lines 802-803, 817-825). Activation of nucleotide metabolism seems to be due to catabolism of nucleic acids and nucleotides. We added clarification (lines 701-705) and reference on research in humans [66]. About 2,3-butanediol we already wrote that it could be both, of microbial (line 843) or plant (lines 849-853) origin. In plants it is detected not only in roots but in leaves (needles) [82]. We are agreed with honorable reviewer that its accumulation and potential role should be further explored, but our findings reported its presents in plant tissues under anoxia.
Comments 5: A more in–depth discussion of the reasons for such divergence and its physiological consequences is required. Whether these differences are attributable to inherent tissue metabolism or varying degrees of anoxia exposure needs to be addressed.
Response 5:
The degree of anoxia effect was similar because according to the experiment designed the entire seedling was placed in anaerobic environment. Thus we believe that revealed differences are linked to tissue/organ sensitivity. Earlier we obtained data with electrolytes leakage test that roots were less damaged by anoxia than shoots of wheat seedlings. We mentioned this in the manuscript text (lines 916-919). Lipids in wheat shoots also damaged greater than in roots by post-anoxic reoxygenation according to TBA-test [98]. Actually different tolerance to oxygen deprivation was demonstrated for roots and shoot of Arabidopsis too [Ellis et al., 1999; doi: 10.1104/pp.119.1.57 ].
Comments 6: The discussion could be strengthened by linking these metabolic irreversibilities to physiological outcomes (e.g., oxidative damage, cell death).Are there any metabolic markers that can predict recovery potential?
Response 6:
At that moment it is very complicated to make final conclusion about metabolic irreversibility in whole scheme of phisiological response. We only could rely on limited literature data of more intensive oxidative damage appearance in shoots [98]. Thus we could not give evidentiary list of metabolic markers. It needs further investigation.
Comments 7: The manuscript is generally well-written, yet some sections are overly complex and would benefit from simplification. For example, the description of metabolite dynamics in Sections 2.3 and 2.5 could be more concise. Although abbreviations (e.g., OPLS - DA, MSEA) are well - defined, their repeated use can impede readability.
Response 7:
We agree with honorable reviewer that all used mathematical method are well-defined and we provided abbreviation list in the end of manuscript. Use of full unabbreviated name would increase the already large text of the article.
Comments 8: The conclusion summarizes the main findings but could more effectively emphasize the broader relevance of the study. For instance, how can these metabolic markers be utilized in breeding or biotechnological approaches to enhance hypoxia tolerance in wheat? Are the identified metabolites conserved across other cereals or sensitive species?
Response 8:
We thank honorable reviewer for this important suggestion. In connection with this, we restructured Conclusion. We took away descriptive part and condensed our finding. We believe that the continuation is needed to evaluate an assumption weather revealed metabolites are conservative across other cereals and non-cereal plants. Now we are providing similar metabolic profiling with rice and preliminary data indicated that rice seedlings accumulated more amino acids and lipids and less amounts hydroxyl carboxylic acids. Thus we are very optimistic for continuation of study of this metabolites for their functional role discover.
Sincerely yours,
Authors
Reviewer 3 Report
Comments and Suggestions for Authors
The present MS focuses on anoxia-induced metabolic changes and adaptation mechanisms at the metabolite level in wheat, a plant species that is less studied under oxygen-deficient conditions, and the results available in the literature and the results also depend on the genotypes and the tested temperature. According to these, the topic of the MS is interesting and the results are novel. Mainly primary metabolites were detected by GC-MS, which is an optimal method for amino acid, TCA cycle compounds, sugars, etc analysis.
The introduction, despite the limited literature, is detailed and highlights the similarities in the available results and contradictions. The authors hypothesize that anoxia-induced metabolite changes will be different at the different sampling points (from 1h to 72 hours). In addition, here a special interest is expected as post-anoxic reoxygenation is also investigated. The tissue-dependent metabolic adaptation is also a special question in this experiment. Such a comparison of short and relatively long periods of anoxia stress, completed with a recovery period, is gap-filling.
Although first the MS seems to be descriptive, different models and analyses (PCA, OPLS, MSEA) help to make a deeper discussion of the results.
The experimental design is correct, the presentation is good, the English is good, easy to follow and understandable. The conclusion is supported by the results and synthesized enough. However, the discussion after the detailed results section is too diffuse (10 and half pages). It is divided into two subsections, one for anoxia and the second for recovery, which is correct, and I found that the authors properly try to discuss their results with those of others, and for example explain the function of certain compounds, but please try to shorten it, especially the first subsection. For example, both subsections end with a last paragraph " Taking together...." are these necessary here? Try to combine them with the conclusion, and avoid duplication.
Other minor corrections are needed:
The sentence in Abstract 30-31 line should be rewritten.
The legend of Fig 2 should be completed to be more specific. Positive loading: means.... Negative loading means..., "Colors in data labels mark chemical class ", e.g. blue are sugars, light green are amino acids, etc. (it is important as later on on Fig 4 this color code will also be used), max -min scale is normalisad to what? 0, 1 and 3 for what?
Fig 3 "a" legend also needs more explanation. These are well described in Suppl Fig 1. legend, but not here. (legend for Fig 3 "b" it is ok")
Fig. 4 "compsug RI 2535 at 6 h, a white circle with asterics? Specify again circle, triangle, etc.
image resolution for suppl fig 2 and 3 not enough good to evaluate.
More info needs about cv. Leningradka, why it was chosen, compared to other wheat genotypes the level of overall stress tolerance?
More info about sampling!
Solve the abbrev: BSTFA : TMCS (for derivatisation)
Author Response
Dear Reviewer!
We are deeply grateful to honorable reviewer for the interest to our results and for very useful comments. We made changes in texts and in figures. Corrections of the text are high lightened in green in the pdf-file. We hope that a new revised version of our manuscript now is better prepared for publication.
Comments and responses:
Comments 1: However, the discussion after the detailed results section is too diffuse (10 and half pages). It is divided into two subsections, one for anoxia and the second for recovery, which is correct, and I found that the authors properly try to discuss their results with those of others, and for example explain the function of certain compounds, but please try to shorten it, especially the first subsection. For example, both subsections end with a last paragraph " Taking together...." are these necessary here? Try to combine them with the conclusion, and avoid duplication.
Response 1:
We tried to rearrange subsection 3.1, we decreased number of paragraphs, added additional subsections and subtitles to make more convenient for readers. But we would like to keep information about the functions of found compounds even if it was elucidated in non-plant organisms. We believe that those compounds are of a special interest for metabolic adaptation and knowing their mechanisms of action in animals and microbes helps to distinguish their function in plants.
Comments 2: The sentence in Abstract 30-31 line should be rewritten.
Response 2:
Corrected
Comments 3: The legend of Fig 2 should be completed to be more specific. Positive loading: means.... Negative loading means..., "Colors in data labels mark chemical class ", e.g. blue are sugars, light green are amino acids, etc. (it is important as later on on Fig 4 this color code will also be used), max -min scale is normalisad to what? 0, 1 and 3 for what?
Response 3:
Figure 2 has a new heatmap legend with numbers, a color key, and an expanded figure caption. Keys have been added for time intervals 0, 1, and 3. Figure 4 were redesigned too, heatmap was altered as large as possible, and stars were colored in bright green for easier visibility. A legend was added, and connection classes named and colored as well.
Comments 4: Fig 3 "a" legend also needs more explanation. These are well described in Suppl Fig 1. legend, but not here. (legend for Fig 3 "b" it is ok")
Response 4:
Symbol explanation and arrows are added to make it easier to understand the direction of changes, and the Figure description expanded. Legend now is OK.
Comments 5: Fig. 4 "compsug RI 2535 at 6 h, a white circle with asterics? Specify again circle, triangle, etc.
Response 5:
Corrected. Key for circle, triangle, etc. is added.
Comments 6: Image resolution for suppl fig 2 and 3 not enough good to evaluate.
Response 6:
We used resolution (600 dpi) that even higher than requirements according to MDPI author’s rules. When combining images into a PDF, resolution appears to be lost. We may provide original FiguresS2&3 to upload into supplementary materials.
Comments 7: More info needs about cv. Leningradka, why it was chosen, compared to other wheat genotypes the level of overall stress tolerance?
Response 7:
Leningradka was the only variety of spring soft wheat zoned for the Leningrad region, Russia. Our region is flood-prone, mainly in spring and early summer. Zonation was cancelled late 2010s. It has been used as intolerant for hypoxia plant in our experiments since early 1980s. We studied different aspects of plant damage and adaptation mechanisms in wheat, barley, oats and rice (reviewed in [2]). Leningradka is totally intolerant to oxygen lack particularly in compare to oats and rice. Seeds for experiment are provided by N.I. Vavilov All-Russian Institute of Plant Genetic Resources, St. Petersburg, Russia.
Comments 8: More info about sampling!
Response 8:
We used randomized mixture of entire roots or shoots from 20 seedlings in 4-8 biological repetitions. Information about sampling is added (lines 1054-1057).
Comments 9: Solve the abbrev: BSTFA : TMCS (for derivatisation)
Response 9:
BSTFA : TMCS is bis(trimethylsilyl)-N,O-trifluoroacetamide : trimethylchlorosilane (line 1068).
Sincerely yours,
Authors
Reviewer 4 Report
Comments and Suggestions for Authors
In general, the study is relevant, and the obtained results have scientific and practical importance. The findings provide valuable information on metabolic profiling of wheat seedlings under anoxia and subsequent reoxygenation. The paper will be of interest to readers.
Here is some suggestion to the quality of the paper presentation:
Abstracts – It should be added the first sentence about the global problem and question addressed in a broad context before highlighting the purpose of the study.
Material and Methods - Lines 1005-1006, please provide information if the plant were germinated and grown in the dark? or under light? (if under light, what photoperiod and light intensity?)
Conclusion – this section could be shortened for better perception by readers, since the Result and Discussion sections already have sufficient information.
The entire text should be double-checked for errors and formatted according to the journal's requirements.
Author Response
Dear Reviewer!
We are would like to tank honorable reviewer for such a positive attitude to our manuscript and for made comments. We used them for revision of Abstract, Material and Methods and Conclusion. At this point we believe that a new revised version of our manuscript is better organized and reader friendly.
Comments and responses:
Comments 1: Abstracts – It should be added the first sentence about the global problem and question addressed in a broad context before highlighting the purpose of the study.
Response 1:
Added, lines 21-22.
Comments 2: Material and Methods - Lines 1005-1006, please provide information if the plant were germinated and grown in the dark? or under light? (if under light, what photoperiod and light intensity?)
Response 2:
Plants were germinated in the dark up to 3 days afterwards they were replanted in hydroponic system and grown at an irradiance of 60 μmol * m-2 * s -1 with a photoperiod of 14/10 h at 23°C. We added required information (line 1040). It is written in cited references [90,101].
Comments 3: Conclusion – this section could be shortened for better perception by readers, since the Result and Discussion sections already have sufficient information.
Response 3:
We shortened the “Conclusion”, excluded references and moved the debatable part into “Discussion”.
Comments 4: The entire text should be double-checked for errors and formatted according to the journal's requirements.
Response 4:
Done
Sincerely yours,
Authors
Round 2
Reviewer 1 Report
Comments and Suggestions for Authors
This manuscript has undergone good revisions, but there are still some areas that can be improved.
- Line 32
Regarding the plant organ with the most obvious metabolome changes after short-term hypoxia, the author stated it was the root in the "Author's Notes", but in the revised manuscript, it was written as the shoots. Please verify which one is correct.
- Previous suggestion 4
Regarding the discussion of the role of these compounds in the life activities of non-plant organisms, it is suggested that the reasons and significance for doing so be added at appropriate positions. For instance, just as the author states in the "Author's Notes".
- Line 1056-1057
The author mentioned here that the sampling site includes the leaves. However, there is no data on leaf metabolomics elsewhere in the manuscript. Please verify if the writing here is correct.
Author Response
We thank honorable reviewer for the overall positive perception of our research and the valuable critique, which was taken into account to improve the manuscript. All suggestions and comments were addressed. Again, corrections in the text are high lightened in green in the pdf-file.
Comments 1: Line 32. Regarding the plant organ with the most obvious metabolome changes after short-term hypoxia, the author stated it was the root in the "Author's Notes", but in the revised manuscript, it was written as the shoots. Please verify which one is correct.
Response 1:
We thank the honorable reviewer for the critical comment and apologize for incorrect statement. The correct one is “root” (line 32)
Comments 2: Previous suggestion 4. Regarding the discussion of the role of these compounds in the life activities of non-plant organisms, it is suggested that the reasons and significance for doing so be added at appropriate positions. For instance, just as the author states in the "Author's Notes".
Response 2:
We added required sentences at appropriate position (lines 577-582).
Comments 3: Line 1056-1057. The author mentioned here that the sampling site includes the leaves. However, there is no data on leaf metabolomics elsewhere in the manuscript. Please verify if the writing here is correct.
Response 3:
Everything is correct. We used shoots (aboveground parts of the seedling) for analysis. 7-day-old wheat seedling possesses two true leaves and short stem (about 3-5 cm) hidden in the leaf sheaths. We used the whole shoot, without separating it into stem and leaves. Proper information is added in Material and Methods (line 1059-1062).
Reviewer 2 Report
Comments and Suggestions for Authors
The revised manuscript has been improved significantly and addresses the majority of my previous comments. I recommend the manuscript for publication
Author Response
We thank the honorable reviewer for the positive perception of our research and useful comments on our manuscript.
SIncerely,
Authors
Round 3
Reviewer 1 Report
Comments and Suggestions for Authors
I have no more opinions.